# Distinguished In Uniform:
# Self-Attention Vs. Virtual Nodes

**Eran Rosenbluth[1]\*, Jan Tönshoff[1]\*, Martin Ritzert[2]\*, Berke Kisin[1] & Martin Grohe[1]**
[1]RWTH Aachen University,  [2]Georg-August-Universität Göttingen,  \* Equal Contribution
{rosenbluth,toenshoff}@informatik.rwth-aachen.de

## Abstract

Graph Transformers (GTs) such as SAN and GPS are graph processing models that combine Message-Passing GNNs (MPGNNs) with global Self-Attention. They were shown to be universal function approximators, with two reservations: 1. The initial node features must be augmented with certain positional encodings. 2. The approximation is non-uniform: Graphs of different sizes may require a different approximating network. We first clarify that this form of universality is not unique to GTs: Using the same positional encodings, also pure MPGNNs and even 2-layer MLPs are non-uniform universal approximators. We then consider uniform expressivity: The target function is to be approximated by a single network for graphs of all sizes. There, we compare GTs to the more efficient MPGNN + Virtual Node architecture. The essential difference between the two model definitions is in their global computation method – Self-Attention Vs Virtual Node. We prove that none of the models is a uniform-universal approximator, before proving our main result: Neither model's uniform expressivity subsumes the other's. We demonstrate the theory with experiments on synthetic data. We further augment our study with real-world datasets, observing mixed results which indicate no clear ranking in practice as well.

## 1 Introduction

In the field of graph learning, message-passing GNNs have long been the undisputed model architecture, even though its basic form is upper bounded in expressivity by the 1-dimensional Weisfeiler-Leman algorithm (Morris et al., 2020; Xu et al., 2019). Recently, transformer architectures, prevalent in language (Vaswani et al., 2017) and vision (Dosovitskiy et al., 2021), shook up the graph learning community (Dwivedi & Bresson, 2020; Rampášek et al., 2022). In a graph transformer, every node is considered a token and then softmax attention is applied between every pair of tokens. This way, graph transformers sidestep the graph structure, enabling them to exploit long-range interactions, a well-known weakness of MPGNNs. Within the message-passing framework, virtual nodes (VNs) represent an alternative solution to incorporate long-range interactions (Gilmer et al., 2017). Understanding the strengths and weaknesses of either approach remains an open problem and is the main focus of this work.

Shortly after the emergence of graph transformers, the expressiveness of such architectures became a natural subject of study with early work by Kreuzer et al. (2021). Like most theoretical papers in graph machine learning (including e.g. Morris et al., 2020; Xu et al., 2019; Abboud et al., 2021), the analysis uses the non-uniform setting where the size and structure of the neural networks may depend on the size of the input graphs. Kreuzer et al. (2021) prove that graph transformers can approximate every function when provided with precomputed positional encodings such as LapPE (Dwivedi & Bresson, 2020). This makes graph transformers seemingly much more expressive than message-passing GNNs that are bounded by 1-WL. We argue that it is not the transformer architecture, but rather the combination of positional encoding and non-uniformity that enable universal function approximation and show that the same result holds for shallow MPGNNs and even MLPs. All three architectures are therefore equally expressive in a non-uniform setting when given sufficiently powerful PEs. We then proceed to compare the models in the uniform setting where a single network must work for graphs of all sizes. The justification for exploring the uniform setting is two-fold:

1) From a theoretical standpoint, uniform expressivity is a significantly stronger notion.
2) Non-uniform expressivity only guarantees the existence of a model that works well for graphs up to a fixed size, while uniform expressivity implies the existence of a model that generalizes to out-of-distribution graph-sizes.

We show that uniform universal approximation is impossible, both for GTs and MPGNN+VNs. We then focus on the differences between the architectures and show that neither can emulate the other, by providing functions that can be expressed by GTs but not by MPGNN+VNs and vice versa. These results focus on model architecture and hold regardless of whether positional encodings like LapPE are used or not. Our theory shows that due to softmax-attention eventually being a weighted average, graph transformers cannot uniformly solve tasks that require unbounded aggregation. In contrast, MPGNNs with virtual nodes that internally perform sum aggregation over all nodes can by definition sum over the nodes in a graph. Based on this observation, we show that the function $|V|^2$ differentiates MPGNNs from GTs. Inversely, we show that a virtual node is not sufficient to emulate the intricate computation of self-attention uniformly, and provide a function that GTs can express but MPGNNs with virtual nodes cannot. A fundamental difference between self-attention and VNs is that in self-attention the updating node can access each of the other nodes individually while with VNs it can only access their aggregated value. However, as proven by Grohe & Rosenbluth (2024), that difference alone does not imply an expressivity advantage. Proving that neither MPGNN+VN nor GT subsumes the other is the main technical contribution of this paper.

We complement our theoretical findings with an empirical study on both synthetic and real-world datasets and observe that a uniformly-expressive model can (sometimes) be learned, and a uniform-inexpressive model does not generalize in terms of graph size. On real-world datasets, we observe mixed results when comparing MPGNNs+VN and GTs, with neither architecture clearly outperforming the other. This loosely aligns with our main result of both architectures being incomparable.

The paper is structured as follows: In Section 2 we introduce concepts that we work with in the following theoretical sections Section 3 and Section 4 where we show the non-uniform and uniform expressivity results. Finally, in Section 5 we provide both the synthetic and real-world experiments.

## 1.1 RELATED WORK

We are not the first to explore the difference between graph transformers and message-passing GNNs with virtual nodes. Cai et al. (2023) prove that with only constant blowup in depth, MPGNN+VN can approximate linear GTs such as Performer. They also show that with a constant number of very wide (i.e. $O(n^d)$) layers, MPGNN+VN can approximate a full self-attention layer. Furthermore, they present an explicit construction using $O(n)$ MPGNN+VN layers of constant width to simulate a self-attention layer, albeit with strong assumptions on the node features. Our theoretical results are perpendicular to theirs. While Cai et al. (2023) aim to mimic one architecture with the other in the non-uniform setting, we analyze the architectures in the uniform setting. We note that their first result about linear graph transformers also holds in the uniform setting, implying that MPGNN+VN architectures subsume linear graph transformers.

Major works in function approximation of transformers are (Yun et al., 2020) for general transformers and (Kreuzer et al., 2021) for graph transformers. Both prove universal function approximation in a non-uniform setting. Kim et al. (2022) show that using (orthogonal) random node identifiers and encoding both nodes and edges as tokens, a standard transformer subsumes 2-WL and thus most pure MPGNN architectures. Note that MPGNNs with random node identifiers are universal approximators for probabilistic functions on graphs (Abboud et al., 2021).

There are a few works that analyze expressiveness in the uniform setting. In Barceló et al. (2020) and Barceló et al. (2021) they characterize the expressive power of GNNs in terms of logic and in the second paper take homomorphism counts as structural encodings into account. Rosenbluth et al. (2023) show that sum-aggregation GNNs do not subsume other aggregations in the uniform setting – as opposed to the non-uniform setting where sum-aggregation GNNs can express anything a GNN can express. Morris et al. (2023) analyze the VC dimension of GNNs in the uniform setting. Other than that, essentially all theoretical papers (silently) assume the non-uniform setting.

In our theoretical analysis and on the synthetic data we compare MPGNNs to GPS (Rampášek et al., 2022) as a prototypical Graph Transformer that works well in practice. Many variations of Graph

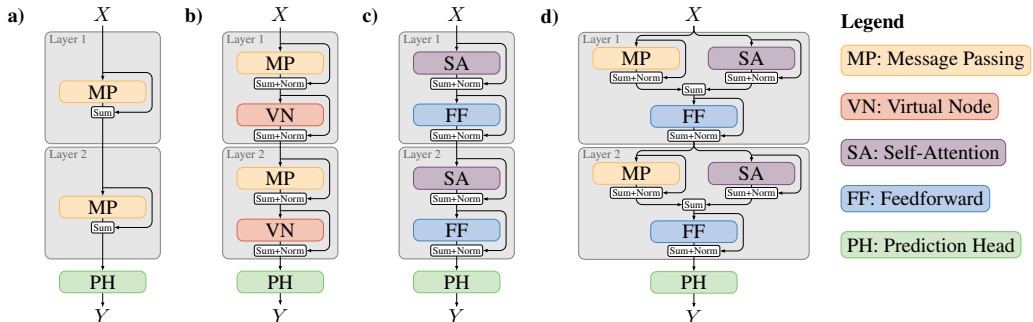

Figure 1: **a)** A stack of MPGNN layers with skip connections **b)** A GNN with a Virtual Node **c)** The Transformer architecture **d)** The GPS architecture as proposed by Rampášek et al. (2022)

Transformers have been proposed and we refer to Müller et al. (2023) for a comprehensive overview. In our real-world experiments, we extend the MPGNN architectures GCN (Kipf & Welling, 2017), GIN/GINE (Xu et al., 2019; Hu et al., 2020b), and GatedGCN (Bresson & Laurent, 2017) with virtual nodes. In addition to GPS, we compare to the graph transformers SAN (Kreuzer et al., 2021), LGI-GT (Yin & Zhong, 2023), and Exphormer (Shirzad et al., 2023).

## 2 PRELIMINARIES

By $\mathbb{N}, \mathbb{N}_{>0}, \mathbb{R}$ we denote the sets of nonnegative integers, positive integers, and real numbers, respectively. For $n \in \mathbb{N}_{>0}$, we use the notation $[n] := \{i \in \mathbb{N} \mid 1 \leq i \leq n\}$. For $a, b \in \mathbb{R} : a < b$, we use the notation $[a, b] := \{r \in \mathbb{R} : a \leq r \leq b\}$ by $[a, b]$. For vectors $v = (v_1, \ldots, v_d) \in \mathbb{R}^d$, we let $|v| := \max(|v_i|_{i \in [d]})$. That is, we use the $\ell_\infty$-norm as our standard vector norm. A (vertex) *featured graph* over feature domain $S^d$, for some set $S$ and $d \in \mathbb{N}$, is a tuple $G = \langle V(G), E(G), Z(G) \rangle$ consisting of a set of vertices $V(G)$, a set $E(G) \subseteq \{\{u, v\} \mid u, v \in V(G)\}$ of (undirected) edges, and a *feature map* $Z(G) \colon V(G) \to S^d$ mapping vertices to vectors in $S^d$. For convenience, we assume that the vertices in $V(G)$ are named $1, \ldots, |V(G)|$. Then, we may denote $Z(G)(i)$ by $Z(G)_i$, and in general, for every mapping of the vertices $f(G)$ we may denote $f(G)(i)$ by $f(G)_i$. For a vertex $v \in V(G)$ we denote by $N(v) := \{w \in V(G) \mid \{w, v\} \in E(G)\}$ its *neighborhood* in $G$. Throughout the paper, we denote the ReLU-activated Multi Layer Perceptron model by MLP. A formal definition of the model can be found in Appendix A.1. For convenience, we will define some operations on a feature map $Z(G)$ as matrix operations. When doing so we view the map as a matrix by stacking all vertex features along the first dimension.

**Message Passing** A *message passing gadget* MPG $M = (U, AGG)$ of dimension $d$ consists of a trainable MLP U with I/O dimensions $2d; d$ and a multiset-to-one aggregation function AGG that aggregates $d$-dimensional vectors over $\mathbb{R}$. In this paper we only consider pointwise Sum, Mean, and Max as aggregation functions as they are the most common choices. The MPG computes a function from a featured graph $G$ to a new feature map:

$$M(G)_v := U\big(Z(G)_v, AGG(\{\{Z(G)_u : u \in N_G(v)\}\})\big). \tag{1}$$

A skip connection follows the pattern $X' = f(X) + X$ and empirically facilitates training deep networks. By wrapping an MPG $M$ into a skip connection, we get a *message-passing layer* MP which computes $MP(G)_v := M(G)_v + Z(G)_v$.

In order to perform graph-level predictions, we use a *readout gadget* $R = (F, AGG)$ consisting of an MLP $F$ with I/O dimensions $d; q$ and AGG an aggregation as above. The readout operation maps a feature map $Z(G)$ of a graph $G$ to a $q$-dimensional vector:

$$R\big(Z(G)\big) := F\big(AGG(\{\{Z(G)_u : u \in V(G)\}\})\big).$$

**MPGNN:** By combining multiple message-passing layers and a readout gadget, we get a *Message-Passing Graph Neural Network* (MPGNN) which is illustrated in Figure 1a. An MPGNN of dimension $p; d; q$ is a tuple $N = (P, L_1, \ldots, L_k, R)$ where $P$ is an initial MLP of I/O dimensions $p; d$

(mapping initial features to $d$-dimensional vectors), $L_1, \ldots, L_k$ are $k$ message-passing layers of dimension $d$, and $R$ is a final readout gadget of dimensions $d; q$ performing graph pooling and (graph-level) prediction. For an input graph $G$, we first apply $P$ to have $G^{(0)} := \langle V(G), E(G), P(Z(G)) \rangle$, where $P(Z(G))$ denotes the application of $P$ in separate to each vertex's initial feature. Then, for each $i \in [k]$ we apply $L_i$ to have $G^{(i)} := \langle V(G), E(G), L_i(Z(G^{(i-1)})) \rangle$. Finally, we apply $R$ to have the final output of the network $N(G) := R(Z(G^{(k)}))$.

**Virtual Node:** The concept of a *Virtual Node* (VN) was suggested by Gilmer et al. (2017) to allow for global information exchange in each layer of a GNN, matching the "intermediate global readout" from Barceló et al. (2020). Formally, a *message-passing layer with VN* (MP+V) of dimension $d$ is given as $\mathrm{MPV} = (\mathrm{MP}, R)$ where MP is a message-passing layer (a message-passing gadget wrapped in a skip-connection) of dimension $d$ and $R$ a readout gadget also of dimension $d$. The virtual node then effectively performs an intermediate aggregation after a round of message passing and adds the result to all nodes. Formally, $\mathrm{MPV}(G)_v := R\big(\mathrm{MP}(G)\big) + \mathrm{MP}(G)_v$.

**MPGNN+VN:** A message-passing GNN with virtual node (MPGNN+VN) $N = (P, L_1, \ldots, L_k, R)$ is defined identically to an MPGNN with the only difference that the layers $L_1, \ldots, L_k$ are now MP+V layers that include a virtual node. We illustrate MPGNN+VNs in Figure 1b.

**Transformer Layer:** An *attention head* H is a function parameterized by three matrices $W_Q, W_K, W_V \in \mathbb{R}^{d \times d_h}$, where $d$ is the input dimension and $d_h$ is the hidden dimension. For an input matrix $X \in \mathbb{R}^{n \times d}$ (e.g. the matrix of node features in a graph) the function $H : \mathbb{R}^{n \times d} \to \mathbb{R}^{n \times d_h}$ is defined as

$$\mathrm{H}(X) := \mathrm{softmax}\left( \frac{(X W_Q)(X W_K)^T}{\sqrt{d_h}} \right) X W_V$$

where the softmax function is applied row-wise. A *self-attention module* SA consists of $k$ attention heads $H_1, \ldots, H_k$ and a weight matrix $W_O \in \mathbb{R}^{k d_h \times d}$ and computes the function

$$\mathrm{SA}(X) := [H_1(X), \ldots, H_k(X)] W_O,$$

where $[\cdot]$ denotes row-wise concatenation. A *Transformer layer* $\mathrm{TL} = (\mathrm{SA}, \mathrm{FF})$ consists of a self-attention module SA and an MLP FF and applies to an input matrix $X \in \mathbb{R}^{n \times d}$:

$$Y := \mathrm{norm}_1\big(X + \mathrm{SA}(X)\big), \qquad \text{and} \qquad Z := \mathrm{norm}_2\big(Y + \mathrm{FF}(Y)\big),$$

where $Z \in \mathbb{R}^{n \times d}$ is the output matrix of the Transformer layer (of dimension $d$) and $\mathrm{norm}_1, \mathrm{norm}_2$ are optional normalization layers such as LayerNorm (Ba et al., 2016) or BatchNorm (Ioffe & Szegedy, 2015) and are applied after skip connections.

**Graph Transformer:** A *Graph Transformer* (GT) stacks Transformer layers to iteratively refine a set of embeddings of the vertices which is illustrated in Figure 1c. Formally, a GT of dimension $p, d, q$ is a tuple $N = (P, L_1, \ldots, L_k, R)$ similar to the other networks with the only difference being that all layers $L_1, \ldots, L_k$ are Transformer layers of dimension $d$.

**GPS:** A *GPS Layer* extends the Transformer layer by a message passing module MP (Equation (1)) which is applied in parallel to the self-attention: Formally, a GPS layer $L = (\mathrm{SA}, \mathrm{MP}, \mathrm{FF})$ of dimension $d$ consists of a self-attention module SA and a message-passing layer MP both of dimension $d$ and an MLP $F$ with I/O dimensions $d; d$. It then computes (again in matrix notation):

$$Y := \mathrm{norm}_1\big(X + \mathrm{SA}(X)\big) + \mathrm{norm}_2\big(X + \mathrm{MP}(X)\big) \quad \text{and} \quad Z := \mathrm{norm}_3\big(Y + \mathrm{FF}(Y)\big)$$

Similar to the transformer layer, we transform the feature map $Z(G)$ of the input graph into a matrix $X$ and interpret the computed matrix $Z$ as the resulting feature map.

Finally, a GPS network $N = (P, L_1, \ldots, L_k, R)$ of dimensions $p; d; q$ is given similar to the other networks with the only difference that this time the layers $L_1, \ldots, L_k$ are all GPS layers. We illustrate a GPS network in Figure 1d.

**Positional Encodings** Graph Transformers typically rely on positional or structural encodings (PE/SE) to inject information about graph structure. Rampášek et al. (2022) provide an overview over common PEs and SEs while new effective encodings are an active field of research (Shiv & Quirk, 2019; Ngo et al., 2023). Formally, a *positional encoding* $\pi(G) : V(G) \to \mathbb{R}^k$ is a special feature map assigning a $k$-dimensional vector to each vertex in a graph based on its local and/or

global structure. In practice, PEs are precomputed before training and combined with the graph's original node features through concatenation or addition.

A common choice is LapPE which is based on the eigendecomposition $(\lambda, \Sigma)$ of the graph Laplacian, thus exploiting spectral graph features. In LapPE, the first $m$ eigenvalues $\lambda$ and eigenvectors $\Sigma$ are accumulated through a learnable function ENC such as a Transformer (in Kreuzer et al., 2021) or DeepSet (Zaheer et al., 2017) as used by Rampášek et al. (2022):

$$\pi(G, v) = \text{ENC}(\Sigma, \lambda, v) \tag{2}$$

Let $M_{\pi,G} = \{\{\pi(G, v) : v \in V(G)\}\}$ be the multiset of vectors that $\pi$ assigns to the vertices of $G$. We say that $\pi$ is *injective up to isomorphism and order $n$* if for all graphs $G, H$ of order at most $n$ it holds that:

$$G \not\cong H \implies M_{\pi,G} \neq M_{\pi,H}. \tag{3}$$

That is, two graphs only map to the same multiset of vertex embeddings if they are isomorphic. It has been shown that LapPE as defined by Kreuzer et al. (2021) has this property when considering the first $n$ eigenvalues. We describe this positional encoding formally in the appendix and state why it is not equivariant. Note that the converse of Equation (3) is not required to hold as PEs, such as LapPE, are often *not* equivariant.

## 3 Non-Uniform Function Approximation on Graphs

In this section, we present universality results that are more or less immediate consequences of MLPs being universal function approximators. Nevertheless, we believe that it is important to discuss these results to clarify that the universality stated in the graph transformer literature is not a unique property of transformers. This has also been briefly discussed by Müller et al. (2023).

Based on the the positional encoding LapPE being injective up to isomorphism and order $n$, Kreuzer et al. (2021) proved that there exists a transformer network $T$ that maps the positional encodings of graphs $G, H$ of order at most $n$ to the same output if and only if $G$ and $H$ are isomorphic. This is a consequence of the injectivity of the positional encoding $\pi$ combined with the universality of transformer networks (Yun et al., 2020). Having realized this, it is clear that the existence of a such an "isomorphism network" is not unique to transformer networks, but shared by other sufficiently powerful architectures including MPGNNs.

**Proposition 3.1.** *2-layer MLPs and 1-layer MPGNNs are universal function approximators when given access to a positional encoding $\pi$ that is injective up to isomorphism and order $n$.*

We give formal statements and proofs for both architectures in the appendix. As research on the expressiveness of GNNs has often focused on the power of GNNs to distinguish graphs, we note that universal function approximation and graph isomorphism testing have been shown to be equivalent (Chen et al., 2019). Thus, both 2-layer MLPs and 1-layer MPGNNs with injective positional encodings are able to distinguish all pairs of graphs. This fact does not contradict the well known limitation of MPGNNs to be at most as powerful as the Weisfeiler-Leman algorithm for distinguishing graphs (Morris et al., 2019; Xu et al., 2019). To the contrary, Corollary B.3 aligns with the fact that if the positional encoding $\pi$ is used as an initial coloring for each graph, then Weisfeiler-Leman distinguishes any two non-isomorphic graphs instantly after zero rounds of refinement. The caveat here is that the algorithm also distinguishes isomorphic graphs $G, H$ whenever $M_{\pi,G} \neq M_{\pi,H}$.

Note that the proofs for Proposition 3.1 as well as the proof of the universality of graph transformers (Kreuzer et al., 2021) share the same principle: Essentially they just implement giant look-up tables for finitely many inputs. Hence, they do not provide any deep insights into the relative strengths and weaknesses of Graph Transformers and MPGNNs.

## 4 Uniform Expressivity of GPS and MPGNN+VNs

We begin with formally defining the terminology and notations related to expressivity.

**Expressivity** We denote the set of graphs featured over $S^d$ by $\mathcal{G}_{S^d}$, we define $\mathcal{G}_S := \bigcup_{d \in \mathbb{N}} \mathcal{G}_{S^d}$, and we denote the set of all featured graphs by $\mathcal{G}_*$. The special set of graphs featured over $\{1\}$ is denoted

$\mathcal{G}_1$. We denote the set of all feature maps that map to $S^d$ by $\mathcal{Z}_{S^d}$, we denote $\bigcup_{d \in \mathbb{N}} \mathcal{Z}_{S^d}$ by $\mathcal{Z}_S$, and we denote the set of all feature maps by $\mathcal{Z}_*$. A mapping $f : \mathcal{G}_{S^d} \to \mathcal{Z}_*$ from featured-graphs to new feature maps is called a *feature transformation*.

Let $p, q \in \mathbb{N}$, and a set $S$. Let $F \subseteq \{f : \ f : \mathcal{G}_{S^p} \to \mathcal{Z}_{\mathbb{R}^q}\}$ be a set of feature transformations, and let $h : \mathcal{G}_{S^p} \to \mathcal{Z}_{\mathbb{R}^q}$ be a feature transformation. We say that $F$ *uniformly additively approximates* $h$, denoted by $F \approx h$, if and only if

$$\forall \varepsilon > 0 \ \exists f \in F : \forall G \in \mathcal{G}_{S^p} \ \forall v \in V(G) \ \ |f(G)(v) - h(G)(v)| \leq \varepsilon$$

The essence of uniformity is that a single function "works" for graphs of all sizes, unlike non-uniform approximation which allows having different functions for graphs of different sizes. In this section, approximation always means uniform additive approximation and we use the term "approximates" synonymously with *expresses*. The expressivity of a model then is the set of functions approximable by a realization of that model.

Let $F, H \subseteq \{f : \ f : \mathcal{G}_{S^p} \to \mathcal{Z}_{\mathbb{R}^q}\}$ be sets of feature transformations, we say that $F$ *subsumes* $H$, notated $F \geq H$ if and only if for every $h : \mathcal{G}_{S^p} \to \mathcal{Z}_{\mathbb{R}^q}$ it holds that $H \approx h \Rightarrow F \approx h$ i.e. everything that can be approximated by functions in $H$ can also be approximated by functions in $F$. If the subsumption holds only for graphs featured with a subset $T^p \subset S^p$ we notate it as $F \geq^T H$, if it does not hold *already* for graphs featured with a subset $T^p \subset S^p$ we notate it as $F \ngeq^T H$.

We call a mapping $f : \mathcal{G}_{S^p} \to \mathbb{R}^q$, from featured graphs to $q$-tuples, a *graph embedding*. We use the approximation terms and notations defined above for feature transformations, for graph embeddings as well.

**Prelude** In the following subsections we prove models' inability to uniformly express various functions. We show that the inexpressivity holds even when the models have reasonable PEs at their disposal e.g. LapPE and RWSE. Such PEs are commonly used in practice and have the following properties: They are polynomial-time computable and their range is bounded. The former is assumed in Theorem 4.1 and the latter in Theorem 4.3. Unlike in Section 3, in the uniform setting injectiveness (of PEs) does not guarantee expressivity. Clearly, the proof logic based on memorizing all inputs is not applicable in an infinite input-domain setting. In addition, commonly used PEs like LapPE and RWSE are not injective for input graphs of unbounded size.

Note that throughout the section we consider a GPS model whose transformer modules incorporate neither BatchNorm nor LayerNorm (see preliminaries). BatchNorm does not increase GPS expressivity, and Layer-Norm would not help GPS in the tasks we define, hence, the lack of them does not affect our inexpressivity results concerning GPS.

The complete proofs of the lemmas and theorems in this section can be found in Appendix C.

## 4.1 GPS AND MPGNN+VNS ARE NOT UNIVERSAL APPROXIMATORS

First, we prove that in the uniform setting, none of the models we consider is a universal approximator.

**Theorem 4.1.** *Let $h$ be the characteristic function of an NP-hard problem on graphs, for example: $h(G) = 1$ if $G$ is 3-colorable and $h(G) = 0$ otherwise. Let Enc: $\mathcal{G}_* \to \mathcal{Z}_*$ a positional encoding function that is computable in polynomial time. Then, assuming PTIME $\neq$ NPTIME we have that GPS $\napprox h$ and MPGNN+VNs $\napprox h$ even when the input graph is featured with positional encodings computed by Enc.*

**Corollary 4.2.** *GPS and MPGNN+VNs cannot approximate every computable function on graphs, not even every graph-function in NPTIME, even when the input graph is featured with PTIME-computable positional encodings.*

## 4.2 GPS DO NOT SUBSUME MPGNN+VNS

Next, we prove that there is a target function that is expressible by a certain MPGNN+VN and is not expressible by any GPS even when the input graphs to the GPS include bounded PEs.

Let Enc: $\mathcal{G}_* \to \mathcal{Z}_{[-1,1]^d}$ be a positional encoding function with bounded range $[-1, 1]^d$ for some $d \in \mathbb{N}$. Define a parameterized, no edges, graph $G_n, n \in \mathbb{N}_{>0}$, featured with PEs according to Enc: $V(G_n) = \{v_1, \ldots, v_n\}; E(G_n) = \emptyset; \forall v \in V(G_n) \ Z(G_n)(v) := \text{Enc}(G_n)(v)$.

**Theorem 4.3.** *Let $f : \mathcal{G}_1 \to \mathbb{R}$ a graph embedding such that for every $n$ it holds that $f(G_n) = n^2$. Then, no GPS can approximate $f$ for all graphs $\{G_n\}$. Formally, GPS $\not\approx f$.*

The proof's main argument is that for every GPS $B$, if the input graphs are of bounded degree and bounded features, then the output of $B$ is bounded – no matter the input graph's size, unlike $f(G_n)$. Combining Theorem 4.3 with the description of an MPGNN+VN that computes $f$ exactly, we arrive at our main conclusion.

**Corollary 4.4.** *GPS $\not\succeq^{[-1,1]}$ MPGNN+VNs.*

### 4.3 MPGNN+VNs Do Not Subsume GPS

Lastly, we prove that there is a target function that is expressible by a certain GPS and is inexpressible by any MPGNN+VN. We define a parameterized, no edges, featured graph $G_{l,r}$, $l, r \in \mathbb{N}_{>0}$, as follows: $V(G_{l,r}) = \{u_1, \ldots, u_l, w_1, \ldots, w_{l\cdot r}\}$; $E(G_{l,r}) = \emptyset$; $Z(G_{l,r}) = \{(u_i, (2,1))\}_{i \in [l]} \cup \{(w_i, (2,2))\}_{i \in [l \cdot r]}$.

Note that all $u_i$ are identical and all $w_j$ are identical, hence we can omit the index when referring to a vertex. Let an MPGNN+VN $B = (P, B_1, \ldots, B_m, R)$ of dimensions $p; d; q$, where $B_i = (M_i, R_i)$ is an MP+V. We first characterize the functions that $B$ can compute for $G_{l,r}$.

**Lemma 4.5.** *There exists a finite set of functions $F = \{f_1, \ldots, f_k\}$, $f_i : \mathbb{N} \times \mathbb{N} \to \mathbb{R}$, such that:*

1. $\forall f \in F \ \ f(l,r) = \sum\limits_{i=0}^{m+1} \sum\limits_{j=0}^{i} \sum\limits_{x=0}^{m+1} \sum\limits_{y=0}^{x} a_{i,j,x,y}^{(f)} l^i r^j \frac{r^y}{(1+r)^x}$ *for some real coefficients $\{a_{i,j,x,y}^{(f)}\}$*

2. $\forall l \in \mathbb{N} \ \forall r \in \mathbb{N} \ \forall j \in [d] \ \exists f, g \in F : Z(G_{l,r}^{(m)})(u)_j = f(l,r)$ *and* $Z(G_{l,r}^{(m)})(w)_j = g(l,r)$.

3. $\forall l \in \mathbb{N} \ \forall r \in \mathbb{N} \ \forall j \in [q] \ \exists f \in F : B(G_{l,r})_j = f(l,r)$

Next, we define a target function and characterize the gap that necessarily exists between it and any function of the kind described in Lemma 4.5 (1).

**Lemma 4.6.** *For $l \in \mathbb{N}_{>0}, r \in \mathbb{N}_{>0}$, define $h(G_{l,r}) := l\big(\frac{3+2re^9}{1+re^9} + r\frac{3+2re^{12}}{1+re^{12}}\big)$.*
*Let $f(l,r) = \sum\limits_{i=0}^{m+1} \sum\limits_{j=0}^{i} \sum\limits_{x=0}^{m+1} \sum\limits_{y=0}^{x} a_{i,j,x,y} l^i r^j \frac{r^y}{(1+r)^x}$ for some coefficients $\{a_{i,j,x,y}\}$, then*

$$\exists r_0 : \forall r > r_0 \ \lim_{l \to \infty} |f(l,r) - h(G_{l,r})| = \infty$$

Combining Lemma 4.5 and Lemma 4.6, we have that the target function is inexpressible by any MPGNN+VN. For $l \in \mathbb{N}_{>0}, r \in \mathbb{N}_{>0}$, define $h(G_{l,r}) := l\big(\frac{3+2re^9}{1+re^9} + r\frac{3+2re^{12}}{1+re^{12}}\big)$.

**Theorem 4.7.** *No MPGNN+VN can approximate $h$ for all graphs $\{G_{l,r}\}$. Formally, MPGNN+VNs $\not\approx h$.*

However, this is not the case for GPS.

**Lemma 4.8.** *There exists a GPS that computes $h$ exactly.*

Combining Theorem 4.7 and Lemma 4.8, we arrive at our main conclusion.

**Corollary 4.9.** *MPGNN+VNs $\not\succeq^{\{1,2\}}$ GPS.*

## 5 Experiments

We perform two kinds of experiments[1]. First, we empirically verify the difference between GPS and MPGNN-VN architectures using synthetic datasets based on Theorems 4.3 and 4.7. Even when an architecture can theoretically express a target function uniformly, it is not guaranteed that SGD-based optimization can learn an expressing model from data. To investigate this, our synthetic experiments test if a uniformly expressive architecture actually yields models with improved o.o.d. generalization to larger graph sizes after training.

---

[1] https://github.com/toenshoff/VN-vs-GT

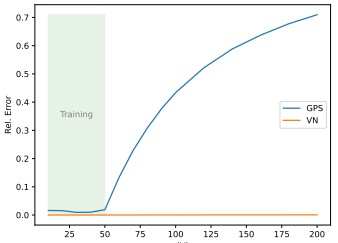 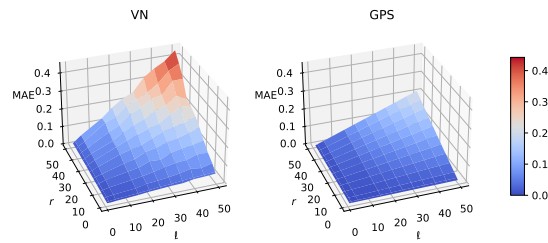

(a) Test performance of MPGNN+VN and GPS when computing the square function. We report the relative error. The graphs in the training data were restricted to $|V| \leq 50$.

(b) Test performance of MPGNN+VN and GPS when computing the function $h(G_{l,r})$ from Section 4.3. The training data is restricted to $l \in [1, 10], r \in [1, 5]$, where $l$ is the number of type-1 vertices and $l \cdot r$ is the number of type-2 vertices.

Figure 2: Synthetic experiments confirming the theoretical differences between models in practice

Second, we perform experiments on real-world datasets and show that neither architecture strictly outperforms the other overall. On some datasets, a simple virtual node is enough to achieve state-of-the-art performance when compared to graph transformers. While these experiments are more loosely related to our theoretical results, they offer another view leading to the general message: Neither GPS nor MPGNN+VN performs strictly better than the other across all learning tasks and a broad range of architectures should generally be considered.

## 5.1 SYNTHETIC EXPERIMENTS

**MPGNN+VN $\not\leq$ GPS** From Corollary 4.4 we know that the simple function where the value $|V|^2$ has to be predicted as a graph-level regression target can be uniformly expressed by an MPGNN with VN but not by a Graph Transformer. In order to verify this empirically, we designed a dataset consisting of 100k random graphs with size $10 \leq |V| \leq 50$. For testing, we use larger graphs of size up to 200 as we are interested in the generalization behavior of the trained models. We train MPGNN+VN and GPS architectures that are identical in width (256 hidden dimensions) and depth (3 layers). For both models we use a standard GCN layer (Kipf & Welling, 2017) as a message passing module. To the GPS model, we additionally provide LapPE encodings. Figure 2a shows the results on the test data. MPGNN+VN has learned to perfectly predict the target function even for graphs outside the training distribution. GPS does achieve a low relative error for graphs with size at most 50. However, for sizes unseen during training the performance deteriorates rapidly. Thus, GPS was unable to learn a function that generalizes across graph sizes which is in line with Corollary 4.4.

**GPS $\not\leq$ MPGNN+VN** To empirically study the converse scenario, we generate another synthetic dataset consisting of 100k random graphs $G_{l,r}$ as defined in the beginning of Section 4.3 with the target function being the graph-level regression target $h(G_l, r)$ from Section 4.3. During training, we restrict the data distribution to $l \in [1, 10]$ and $r \in [1, 5]$ and increase the range to $l, r \in [1, 50]$ for testing. Again, we train 3-layer MPGNN+VN and GPS models based on GCN layers of identical width. Figure 2b provides the experiment's results. For both models the MAE increases roughly linearly in both $l$ and $r$. The error of the MPGNN+VN model increases around twice as quickly as that of the GPS model which generalizes better to values of $r$ and $l$ outside the training distribution which aligns with Corollary 4.9.

Through the synthetic experiments, we showed that the theoretical findings from Section 4 extend to practice. Especially in the first case, we observe that a simple function on the number of nodes of a graph was not learned robustly by GPS but was learned by the uniformly expressive MPGNN+VN architecture.

## 5.2 EXPERIMENTS ON BENCHMARK DATASETS

Finally, we compare MPGNN+VNs to Graph Transformers to contrast the two paradigms of global information exchange. For this, we evaluate MPGNN+VNs on real-world benchmark datasets from both the recent LRGB collection (Dwivedi et al., 2022) and OGB (Hu et al., 2020a). Concretely, we

Table 1: Performance measurements of VN-equipped models. **First**, **second**, and **third** best values are colored. * indicates methods lacking feature normalization on the computer vision datasets.

| | METHOD | PASCALVOC-SP TEST F1 ↑ | COCO-SP TEST F1 ↑ | PEPTIDES-FUNC TEST AP ↑ | PEPTIDES-STRUCT TEST MAE ↓ | OGB-CODE2 TEST F1 ↑ | OGB-MOLPCBA TEST AP ↑ |
|---|---|---|---|---|---|---|---|
| MPGNN | GCN | 0.2078 ± 0.0031 | 0.1338 ± 0.0007 | 0.6860 ± 0.0050 | 0.2460 ± 0.0007 | 0.1507 ± 0.0018 | 0.2483 ± 0.0037 |
| | GINE / GIN | 0.2718 ± 0.0054 | 0.2125 ± 0.0009 | 0.6621 ± 0.0067 | 0.2473 ± 0.0017 | 0.1495 ± 0.0023 | 0.2703 ± 0.0023 |
| | GATEDGCN | 0.3880 ± 0.0040 | 0.2922 ± 0.0018 | 0.6765 ± 0.0047 | 0.2477 ± 0.0009 | 0.1606 ± 0.0015 | 0.3066 ± 0.0013 |
| GTs | SAN | 0.3230 ± 0.0039* | 0.2592 ± 0.0158* | 0.6439 ± 0.0075 | 0.2545 ± 0.0012 | - | 0.2765 ± 0.0042 |
| | GPS | 0.4440 ± 0.0065 | 0.3884 ± 0.0055 | 0.6534 ± 0.0091 | 0.2509 ± 0.0014 | 0.1894 ± 0.0024 | 0.2907 ± 0.0028 |
| | LGI-GT | - | - | - | - | 0.1948 ± 0.0024 | 0.3040 ± 0.0029 |
| | EXPHORMER | 0.3975 ± 0.0037* | 0.3455 ± 0.0009* | 0.6527 ± 0.0043 | 0.2481 ± 0.0007 | - | - |
| VN | GCN-VN | 0.2950 ± 0.0058 | 0.2072 ± 0.0043 | 0.6732 ± 0.0066 | 0.2505 ± 0.0022 | - | - |
| | GATEDGCN-VN | 0.4477 ± 0.0137 | 0.3244 ± 0.0025 | 0.6823 ± 0.0069 | 0.2475 ± 0.0018 | 0.1855 ± 0.0018 | 0.3141 ± 0.0019 |

use the two graph-level peptides datasets peptides-func and peptides-struct and the two node-level superpixel datasets Pascal-VOC and COCO from LRGB and the two graph-level datasets code2 and MolPCBA from OGB. Details on the datasets are given in the appendix. We note that on LRGB Tönshoff et al. (2023) very recently showed that the original baseline results reported by Dwivedi et al. (2022) and Rampášek et al. (2022) suffered from suboptimal hyperparameters and lack of feature normalization. We thus use their updated baseline results on LRGB.

**Results** Our results are provided in Table 1. We can see that on the peptides datasets, and especially peptides-struct, all models are very close and neither Graph Transformers nor MPGNNs with virtual nodes hold a significant edge over simple MPGNNs. On the superpixel datasets PascalVOC-SP and COCO-SP the performance of MPGNNs is significantly strengthened by adding a virtual node. On PascalVOC-SP the GatedGCN+VN model is on par with GPS, indicating that the virtual node yields similar performance benefits as the self-attention in GPS which also relies on GatedGCN as its message passing module on this dataset. On COCO-SP the improvement of adding a VN to MPGNNs is also significant, although not enough to fully close the gap to GPS. The results on ogbg-code2 are similar and adding virtual nodes to MPGNNs significantly boosts the performance to almost the level of the GTs. Last, we provide new results for GatedGCN on ogbg-molpcba (with and without virtual nodes) and achieved state-of-the-art performance. Here, virtual nodes were able to further boost the already very good performance of GatedGCN by about 1% AP which is a highly significant gain over all other methods and even reaches the level of Graphormer (not shown) which was pretrained on additional data.

Overall, the comparison yields mixed results with neither Graph Transformers nor MPGNNs+VN clearly outperforming the other. This adds to our main theoretical results which state that neither architecture subsumes the other in terms of uniform expressivity. Furthermore, MPGNNs+VN do yield the best results on some datasets, showing that virtual nodes can generally compete with self-attention as a means of global information exchange on real-world learning tasks while being significantly more efficient.

# 6 CONCLUSION

In the non-uniform setting we formally proved that strong positional encodings imply universality not only for GTs but also for MPGNNs and even MLPs. We thus looked into uniform approximation where a single model has to work for all graph sizes. There we proved that GTs and MPGNNs with virtual nodes express different sets of functions. The core observations were that GTs are unable to perform unbounded aggregation (e.g. counting the nodes) and on the other hand self-attention can compute functions that can not be expressed uniformly by the simple computation of a virtual node. Through synthetic experiments, we showed that this difference also extends to practice. On real-world datasets from LRGB and OGB we observe that simple and efficient virtual nodes are generally competitive with graph transformers. To paraphrase this with a popular saying; in graph learning, attention is often not *all* you need.

ACKNOWLEDGMENTS

This work was supported by the German Research Foundation (DFG) under grants GR 1492/16-1 and KI 2348/1-1 "Quantitative Reasoning About Database Queries".
This work is funded by the Deutsche Forschungsgemeinschaft (DFG) – 2236/2 (UnRAVeL).

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

# A ADDITIONAL DEFINITIONS

## A.1 MULTI LAYER PERCEPTRON

A ReLU-activated *Multi Layer Perceptron* (MLP) of I/O dimensions $p; q$, consisting of $m$ layers, is a tuple $M = (W^{(1)} \in \mathbb{R}^{d_1 \times p}, b^{(1)} \in \mathbb{R}^{d_1}, \ldots, W^{(m)} \in \mathbb{R}^{q \times d_{m-1}}, b^{(m)} \in \mathbb{R}^q)$ for some intermediate dimensions $d_1, \ldots, d_{m-1}$. It determines a function from an input vector $X \in \mathbb{R}^p$ to a vector $Y \in \mathbb{R}^q$:

$$M(X) = b^{(m)} + W^{(m)} ReLU(b^{(m-1)} + W^{(m-1)} \cdots ReLU(b^{(1)} + W^{(1)} X))$$

where $ReLU(x) := \max(0, x)$. We consider the size of $M$ to be the total size of its underlying matrices and biases vectors.

In this paper, we assume all MLPs to be ReLU activated. ReLU activated MLPs subsume every finitely-many-pieces piecewise-linear activated MLP, thus the results of this paper hold true for all such MLPs. Every ReLU-activated MLP $M$ is Lipschitz-continuous. That is, there exists a minimal $a_M \in \mathbb{R}_{\geq 0}$ such that for every input and output coordinates $(i, j)$, for every specific input arguments $x_1, \ldots, x_n$, and for every $\delta > 0$, it holds that

$$|M(x_1, \ldots, x_n)_j - M(x_1, \ldots x_{i-1}, x_i + \delta, \ldots, x_n)_j| / \delta \leq a_M$$

We call $a_M$ the *Lipschitz-constant* of $M$.

## A.2 POSITIONAL ENCODINGS

In the following we describe the positional encoding LapPE as defined by Kreuzer et al. (2021) formally. We then describe at the example of LapPE why PEs are in general not equivariant. Note that this lack of equivariance might be detrimental in graph learning as a difference in the PE does typically not imply that the graphs are isomorphic.

**Example A.1** (Kreuzer et al. (2021)). *A standard positional encoding uses a basis of eigenvectors of the Laplacian matrix. Let $G = (V, E)$ be a graph, and let $L_G \in \mathbb{R}^{V \times V}$ be its Laplacian matrix. Recall that $L_G$ has the vertex degrees as diagonal entries and at off-diagonal position $v \neq w$ entry $-1$ if $v, w$ are adjacent and entry $0$ otherwise. $L_G$ is positive semi-definite and has nonnegative eigenvalues $\lambda_1 \leq \lambda_2 \ldots \leq \lambda_n$. Let $\boldsymbol{u}_1, \ldots, \boldsymbol{u}_n$ be an orthonormal basis of corresponding eigenvectors. These vectors are indexed by vertices of $G$, that is, $\boldsymbol{u}_i \in \mathbb{R}^V$, and we let $u_{iv}$ be the $v$-entry of $\boldsymbol{u}_i$. We define a positional encoding $\pi_{Lap}$ by*

$$\pi_{Lap}(G, v) = (\lambda_1, u_{1v}, \lambda_2, u_{2v}, \ldots, \lambda_n, u_{nv})$$

*(since $\lambda_1 = 0$ and $\boldsymbol{u}_1 = \boldsymbol{1}$, we can also omit the first two entries). This positional encoding is injective up to isomorphism, because the graph $G$ can be reconstructed from the eigenvectors $\boldsymbol{u}_i$ and eigenvalues $\lambda_i$.[2] To make this positional encoding practical, but still keep it injective, we can round the reals to about $n$ digits for graphs of order $n$.*

*Note that $\pi_{Lap}$ not only depends on $G$, but also on the choice of the vectors $\boldsymbol{u}_i$, which is not canonical even if we normalize the vectors, because the signs cannot be normalized and eigenspaces may have dimension greater than $1$. Formally, this means that the mapping $\pi$ is not equivariant.*

Note that as described in the paper, the non-equivariance of LapPE in the example shows that the converse of the implication Equation (3) is not required to hold.

---

[2] It is necessary to put the eigenvalues as well as the entries of the vectors in the positional encoding. Neither the eigenvalues alone nor the eigenvectors alone are sufficient to make the encoding injective. For the former, just note that there are well-known examples of nonisomorphic co-spectral graphs (see, e.g., Godsil & Royle, 2001, Section 8.1). For the latter, let $G$ be the graph with two vertices and no edges with Laplacian $L_G = \begin{pmatrix} 0 & 0 \\ 0 & 0 \end{pmatrix}$, and let $H$ be the graph with two vertices and one edge with Laplacian $L_H = \begin{pmatrix} 1 & -1 \\ -1 & 1 \end{pmatrix}$. The eigenvalues of $L_G$ are $\lambda_1 = \lambda_2 = 0$, and the eigenvalues of $L_H$ are $\lambda_1 = 0, \lambda_2 = 2$. However, for both graphs $\boldsymbol{u}_1 = (1/\sqrt{2}, 1/\sqrt{2})^\top, \boldsymbol{u}_2 = (1/\sqrt{2}, -1/\sqrt{2})^\top$ is an orthonormal basis of eigenvectors.

## B   EXPRESSIVITY RESULTS IN THE NON-UNIFORM SETTING

For the remainder of this section, fix some $n \in \mathbb{N}$ and let $\pi$ be some positional encoding that is injective up to isomorphism and order $n$, e.g. LapPE by Kreuzer et al. (2021), see Example A.1 in the appendix.

As stated in the main paper, the universality proofs of graph transformers and other architectures in the non-uniform setting relies on the universality of MLPs. The following proposition extracts the core of the argument. For an $n$-vertex graph $G$ with vertices $v_1, \ldots, v_n$, we let $\pi\big(G, (v_1, \ldots, v_n)\big)$ be the vector of length $nd$ obtained by stacking the positional encodings of all vertices.

**Proposition B.1.** *Let $\pi$ be a positional encoding that is injective up to isomorphism and order $n$. Let $f$ be an arbitrary function that maps graphs to $\mathbb{R}^q$. Then for every $n \geq 1$ and $\varepsilon > 0$ there is a 2-layer MLP $N$ of input dimension $nd$ and output dimension $q$ such that for all graphs $G$ of order $n$ and all enumerations $\boldsymbol{v} = (v_1, \ldots, v_n)$ of the vertices of $G$ we have*

$$\big\| N\big(\pi(G, \boldsymbol{v})\big) - f(G) \big\| \leq \varepsilon.$$

*Proof.* Let $X$ be the set of all $\pi(G, \boldsymbol{v})$, where $G$ is a graph of order $n$ and $\boldsymbol{v} = (v_1, \ldots, v_n)$ an enumeration of its vertices. Note that $X$ is a finite and hence compact subset of $\mathbb{R}^{nd}$. Let $g : \mathbb{R}^{nd} \to \mathbb{R}^q$ be a continuous function such that for all $\boldsymbol{x} \in X$ it holds that $g(\boldsymbol{x}) = f(\boldsymbol{x})$.

By the Universal Approximation Theorem (Cybenko, 1989; Hornik, 1991) for MLPs, there is a 2-layer MLP $N$ that approximates $g$ on the compact set $X$ with an additive error bounded by $\varepsilon$. Then for every graph $G$ of order $n$ and every enumeration $\boldsymbol{v} = (v_1, \ldots, v_n)$ of its vertices, we have

$$\big\| N\big(\pi(G, \boldsymbol{v})\big) - f(G) \big\| \;=\; \big\| N\big(\pi(G, \boldsymbol{v})\big) - g\big(\pi(G, \boldsymbol{v})\big) \big\| \;\leq\; \varepsilon. \qquad \square$$

The following proposition is closely related to (Dasoulas et al., 2020) (though slightly different, because we do not rely on unique node identifiers, but just use multisets of node labels). Let $N(G, \pi)$ denote the result of the computation of an MPGNN $N$ on $G$ with initial vertex states $\pi(G, \boldsymbol{v})$.

**Proposition B.2** (1-layer MPGNNs are "universal"). *Let $\pi$ be a positional encoding that is injective up to isomorphism and order $n$. Let $f$ be an arbitrary function mapping graphs to $\mathbb{R}^q$. Then for every $\varepsilon > 0$ there is a graph-level MPGNN $N$ with a single message passing layer and sum readout such that for all graphs $G$ of order at most $n$ we have*

$$\big\| N(G, \pi) - f(G) \big\| \leq \varepsilon.$$

The argument is similar to the one given in the proof of Proposition B.1, albeit slightly more complicated. The key difference is that the information that is stored in the node encoding must survive the readout step where a sum over all nodes is computed. We thus need to first use another 2-layer MLP that maps the encoding to something that can be safely aggregated through summation without losing information. We can then use the universality of MLPs again to approximate the desired function.

*Proof.* Let $n \geq 1$ and $\ell := \lfloor \log n \rfloor + 1$. Let $X$ be the set of all $\pi(G, v)$, where $G$ is a graph of order $n$ and $v$ a vertex of $G$. Then $X$ is a finite subset of $\mathbb{R}^d$. Suppose that $X = \{x_1, \ldots, x_k\}$. Let $h : \mathbb{R}^d \to \mathbb{R}$ be a continuous function with $h(x_j) = 2^{-j\ell}$ for $j = 1, \ldots, k$, and let $\delta := \frac{1}{n2^{k\ell+2}}$. By the Universal Approximation Theorem, there is a 2-layer MLP $N_h$ that approximates $h$ on the compact set $X$ with an additive error bounded by $\delta$. Then for every graph $G$ with vertex set $v_1, \ldots, v_n$ we have

$$\left| \sum_{i=1}^{n} N_h\big(\pi(G, v_i)\big) - \sum_{i=1}^{n} h\big(\pi(G, v_i)\big) \right| \leq n\delta = 2^{-k\ell-2}$$

Next, we use $h$ to injectively map multisets $M \subseteq X$ of size $n$ to $[0, 1]$: let $M$ be such a multiset, and for every $j \in [k]$, let $m_j \leq n$ be the multiplicity of $x_j$ in $M$. We let

$$H(M) := \sum_{j=1}^{k} m_i 2^{-j\ell}.$$

The mapping is injective, because for $m \leq n < 2^\ell$ and $j \leq k$ it holds that $2^{-j\ell} \leq m2^{-j\ell} < 2^{-(j-1)\ell}$. Furthermore, for distinct multisets $M, M'$ it holds that $|H(M) - H(M')| \geq 2^{-k\ell}$. Observe that for every graph $G$ of order $n$, $M_{\pi,G} \subseteq X$ is a multiset of order at most $n$. If $v_1, \ldots, v_n$ is an enumeration of the vertex set of $G$, we have $H(M_{\pi,G}) = \sum_{i=1}^n h(\pi(G, v_i))$. Thus

$$\left| \sum_{i=1}^n N_h\big(\pi(G, v_i)\big) - H(M_{\pi,G}) \right| \leq 2^{-k\ell-2}.$$

Let $Y$ be the set of all $H(M_{\pi,G})$, where $G$ is a graph of order $n$. Then $Y \subseteq \mathbb{R}$ is a finite set, and for distinct $y, y' \in Y$ it holds that $|y - y'| \geq 2^{-k\ell}$. Let $Z$ be the set of all $z \in \mathbb{R}$ such that $|z - y| \leq 2^{-k\ell-2}$ for some $y \in Y$. Then $Z$ is a compact set, and for every $z \in Z$ there is a unique graph $G_z$ of order $n$ such that $|z - H(M_{\pi,G_z})| \leq 2^{-k\ell-2}$.

Let $g : \mathbb{R} \to \mathbb{R}^q$ be a continuous function such that for all $z \in Z$ it holds that $g(z) = f(G_z)$. By the Universal Approximation Theorem, there is a 2-layer MLP $N_g$ that approximates $g$ on the compact set $Z$ with an additive error bounded by $\varepsilon$. Then for all graphs $G$ with vertices $v_1, \ldots, v_n$, we have

$$\left\| N_g\left( \sum_{i=1}^n N_h\big(\pi(G, v_i)\big) \right) - f(G) \right\| \leq \varepsilon.$$

To see this, let $z = \sum_{i=1}^n N_h\big(\pi(G, v_i)$. Then $|z - H(M_{\pi,G})| \leq 2^{-kl-2}$. Thus $G_z = G$ and hence $|N_g(z) - g(z)| = |N_g(z) - f(G)| \leq \varepsilon$.

It remains to implement the function $(G, \pi) \mapsto N_g\left(\sum_{i=1}^n N_h\big(\pi(G, v_i)\big)\right)$ by a 1-layer MPGNN. The message passing layer simply maps the state $\boldsymbol{x}(v)$ of node $v$ to $N_h(\boldsymbol{x}(v))$, ignoring the messages the node receives from its neighbors. Then global aggregation yields $\sum_{i=1}^n N_h(\boldsymbol{x}(v))$. The readout function is $N_g$. Thus, if the initial state $\boldsymbol{x}(v)$ is $\pi(G, v)$, then this MPGNN computes the desired function. $\qquad\square$

The following corollary formalizes that MPGNNs with injective PEs are able to distinguish all pairs of graphs as discussed in the main paper.

**Corollary B.3.** *Let $\pi$ be a positional encoding that is injective up to isomorphism and order $n$. There exists a graph-level MPGNN $N$ with a single message passing layer and sum readout that maps the encodings of two graphs $G, H$ of order at most $n$ to the same output if and only if $G$ and $H$ are isomorphic:*

$$N(G, \pi) = N(H, \pi) \iff G \cong H$$

As also discussed in the paper, this does not contradict the fact that MPGNNs are bounded by 1-WL as the additional features also strengthen 1-WL to the point that 1-WL also distinguishes all pairs of graphs (and actually also isomorphic graphs for which the PE is not equivariant and the embeddings thus differ).

## C    Uniform Expressivity, Proofs

Note that throughout the section we consider a GPS model whose transformer modules incorporate neither BatchNorm nor LayerNorm (see preliminaries). BatchNorm does not increase GPS expressivity, and Layer-Norm would not help GPS in the tasks we define, hence, the lack of them does not affect our inexpressivity results.

**Theorem 4.1.** *Let $h$ be the characteristic function of an NP-hard problem on graphs, for example: $h(G) = 1$ if $G$ is 3-colorable and $h(G) = 0$ otherwise. Let Enc: $\mathcal{G}_* \to \mathcal{Z}_*$ a positional encoding function that is computable in polynomial time. Then, assuming PTIME $\neq$ NPTIME we have that GPS $\not\approx h$ and MPGNN+VNs $\not\approx h$ even when the input graph is featured with positional encodings computed by Enc.*

*Proof.* Note that in the uniform setting – having one network to handle graphs of all sizes – the algorithm implied by any (GPS or MPGNN+VN) network has a runtime polynomial in the size of the input graph. Assume to the contrary that there exists a (GPS or MPGNN+VN) network $N$ that

approximates $h(G)$ every input graph $G$ featured with $Enc(G)$ as positional encoding. Then, it is possible to approximate $h(G)$ in polynomial time: Compute $Enc(G)$ in polynomial time and run $N$ on $G$ featured with $Enc(G)$. This is in contradiction to $h$ being NP-hard and the assumption that PTIME $\neq$ NPTIME. $\qquad\square$

**Lemma C.1.** *Let $L = (A, M, F)$ be a $p$-dimensional GPS layer such that $A = (W_Q, W_K, W_V, W_O)$ is a $(p; q; p)$-dimensional attention module (for some $q$), $M = (H, AGG)$ is a $p$-dimensional MPG with $AGG \in \{mean, sum, max\}$, and $F$ is a $p; p$-dimensional MLP. Then, if the input features and the graph degree are bounded, then so are the values of the output feature map. Formally, for every $b \in \mathbb{R}, d \in \mathbb{N}$ there exists $b' \in \mathbb{R}$ such that: For every input graph $G$ of degree at most $d$ and for which $\max(|Z(G)_{i,j}| : i \in [|G|], j \in [p]) \leq b$, it holds that $\max(|A(G)_{i,j}| : i \in [|G|], j \in [p]) \leq b'$.*

*Proof.* 1) The linear transformation defined by $W_V$ is Lipschitz-continuous, and so the bounded input features determine a bounded range for the value vectors coordinates. By definition of the attention mechanism, the output of $A$ is no higher than the maximum over the value vectors. Hence, there exists $b'_A$ such that $\max(|A(Z(G))_{i,j}| : i \in [|G|], j \in [r]) \leq b'_A$.

2) An MLP is Lipschitz-continuous, hence for $AGG \in \{mean, \max\}$ we have $M$ to be Lipschitz-continuous, and for a bounded degree input graph we have also for $AGG = sum$ that $M$ is Lipschitz-continuous.

3) The addition and composition of Lipschitz-continuous functions is Lipschitz-continuous and so for a graph-domain that is both degree-bounded and feature-bounded we have that $A$ is bounded. $\qquad\square$

**Theorem 4.3.** *Let $f : \mathcal{G}_1 \to \mathbb{R}$ a graph embedding such that for every $n$ it holds that $f(G_n) = n^2$. Then, no GPS can approximate $f$ for all graphs $\{G_n\}$. Formally, GPS $\not\approx f$.*

*Proof.* Let a GPS $B = (P, B_1, \ldots, B_m, R)$ of dimension $p; d; q$ where $R = (F_R, AGG_R)$. The degree of any $G_n$ is bounded (to be zero), the features are bounded, and since $P$ is Lipschitz-continuous also $P(Z(G))$ (denoting the application of $P$ in separate to each vertex's initial feature) is bounded. Hence, by Lemma C.1 we have that $B_1(G)$ is bounded. By induction, using Lemma C.1, we have that the output of every layer $B_i$ is bounded, denote the bound on the output of $B_m$ by $b_m$. Finally, $R$ is Lipschitz-continuous, denote its Lipschitz-constant by $a_R$, then for $AGG_R \in \{mean, max, sum\}$ we have that $B$ is bounded by $n(p \cdot b_m \cdot d \cdot a_R)$. Then, given $\delta > 0$, we set $n = (\delta + p \cdot b_m \cdot d \cdot a_R + 1)$ and get

$$|f(G_n) - B(G_n)| \geq n(n - p \cdot b_m \cdot d \cdot a_R) \geq n(\delta + 1) > \delta$$

$\qquad\square$

**Corollary 4.4.** *GPS $\not\succeq^{[-1,1]}$ MPGNN+VNs.*

*Proof.* By Theorem 4.3 no GPS+(bounded)PEs can approximate $f$. It is not difficult to verify that a MPGNN+VN with one MPG+V with a sum-readout gadget, and a final sum-readout, can compute $f$ exactly. $\qquad\square$

**Lemma C.2.** *Let a $p; q$-dimensional MLP $M = (W^{(1)} \in \mathbb{R}^{d_1 \times p}, b^{(1)} \in \mathbb{R}^{d_1}, \ldots, W^{(m)} \in \mathbb{R}^{q \times d_{m-1}}, b^{(m)} \in \mathbb{R}^q)$ , that is, for every input vector $X = (x_1, \ldots, x_p) \in \mathbb{R}^p$ it holds that $M(X) = b^{(m)} + W^{(m)} ReLU(b^{(m-1)} + W^{(m-1)} \cdots ReLU(b^{(1)} + W^{(1)} X))$. Then, regardless of the input, every output of $M$ is one of finitely many affine functions. Formally, there exists a finite set $F = \{f_1, \ldots, f_k\}$ of affine functions $f_i : \mathbb{R}^p \to \mathbb{R}, f_i(X) = c_i + \sum_{j=1}^{p} (a_{i,j} X_j)$ such that:*

$$\forall X \in \mathbb{R}^p \; \forall i \in [q] \; \exists f \in F : M(X)_i = f(X)$$

*Proof.* By induction on the number of layers of $M$. For $t = 1$ we have that either $ReLU(b^{(1)} + W^{(1)} X)_j = 0$ or $ReLU(b^{(1)} + W^{(1)} X)_j = b_j^{(1)} + W_{j,.}^{(1)} X$, hence the set $F = \{0, (b_1^{(1)} + W_{1,.}^{(1)} X), \ldots, (b_{d_1}^{(1)} + W_{d_1,.}^{(1)} X)\}$ satisfies the requirement.

Assuming correctness for $t = n - 1$, we prove for $t = n$: Let $F^{(n-1)}$ be the function-set that satisfies the lemma up to layer $n - 1$. Let $Y \in \mathbb{R}^{d_{n-1}}$ be the output of layer $n - 1$, we have that either $ReLU(b^{(n)} + W^{(n)}Y)_j = 0$ or $ReLU(b^{(n)} + W^{(n)}Y)_j = b_j^{(n)} + W_{j,.}^{(n)}Y$, and by assumption $b_j^{(n)} + W_{j,.}^{(n)}Y = b_j^{(n)} + W_{j,.}^{(n)}(g_1(X), \ldots, g_{d_{n-1}}(X))$ for some $g_i \in F^{(n-1)}$. Hence, $ReLU(b^{(n)} + W^{(n)})_j \in \{0\} \bigcup \{b_j^{(n)} + W_{j,.}^{(n)}(g_1(X), \ldots, g_{d-1}(X)) : g_i \in F^{(n-1)}\}$, and since a linear combination of affine functions is an affine function the latter is a finite set of affine functions of $X$. $\qquad\square$

**Lemma 4.5.** *There exists a finite set of functions $F = \{f_1, \ldots, f_k\}$, $f_i : \mathbb{N} \times \mathbb{N} \to \mathbb{R}$, such that:*

1. $\forall f \in F$  $f(l, r) = \sum_{i=0}^{m+1} \sum_{j=0}^{i} \sum_{x=0}^{m+1} \sum_{y=0}^{x} a_{i,j,x,y}^{(f)} l^i r^j \frac{r^y}{(1+r)^x}$ *for some real coefficients $\{a_{i,j,x,y}^{(f)}\}$*

2. $\forall l \in \mathbb{N} \; \forall r \in \mathbb{N} \; \forall j \in [d] \; \exists f, g \in F : Z(G_{l,r}^{(m)})(u)_j = f(l, r) \text{ and } Z(G_{l,r}^{(m)})(w)_j = g(l, r)$.

3. $\forall l \in \mathbb{N} \; \forall r \in \mathbb{N} \; \forall j \in [q] \; \exists f \in F : B(G_{l,r})_j = f(l, r)$

*Proof.* Given $t \in [m], i \in [l], j \in [lr]$, we define shorthand notations for the values of any of the $u_i$ and $w_j$, after the $t^{th}$ MPG+V:

- $u_{l,r}^{(t)} := Z(G_{l,r}^{(t)})(u_i), i \in [l]$

- $w_{l,r}^{(t)} := Z(G_{l,r}^{(t)})(w_j), j \in [lr]$

By induction on the layer of the MPGNN+VN:

a) For $t = 0$, the preparation gadget is an MLP, and so its output is one of finitely many affine functions of the initial vertex values $u_{l,r}^{(0)}, w_{l,r}^{(0)}$.

b) Assume correctness for $t = n - 1$, we prove for $t = n$. Let an MPG+V $B_n = (M_n, R_n)$ where $M_n = (U_n, AGG_n)$ is an MPG and $R_n = (F_n, RO_n)$ is a readout gadget.

b.1 Since there are no edges, $AGG_n(u_{l,r}^{(n-1)}) = 0$, and so $M_n(G_{l,r}, u^{(n-1)}) = U_n(u_{l,r}^{(n-1)})$. By the induction assumption $U_n(u_{l,r}^{(n-1)}) = U_n(g_1(l, r), \ldots, g_q(l, r))$ where $g_i \in F$ for some finite set $F$ of functions of the form $f(l, r) = \sum_{i=0}^{n} \sum_{j=0}^{i} \sum_{x=0}^{n} \sum_{y=0}^{x} a_{i,j,x,y} l^i r^j \frac{r^y}{(1+r)^x}$. By Lemma C.2, for every sequence $(g_1, \ldots, g_q) \in F^q \; \forall j \in [q] \; U_n(g_1(l, r), \ldots, g_q(l, r))_j$ is one of finitely many affine functions of $(g_1, \ldots, g_q)$ and so overall there is a finite set $F'$ of functions of the form $f(l, r) = \sum_{i=0}^{n} \sum_{j=0}^{i} \sum_{x=0}^{n} \sum_{y=0}^{x} a_{i,j,x,y}^{(f)} l^i r^j \frac{r^y}{(1+r)^x}$ such that $\forall j \in [q] \; M_n(G_{l,r}, u_{l,r}^{(n-1)})_j \in F'$. Similarly, $M_n(G_{l,r}, w_{l,r}^{(n-1)})_j \in F''$ for some finite $F''$ of functions of the same form.

b.2 If $RO_n = sum$ then $R_n(M_n(G_{l,r}^{(n-1)})) = F_n(l(M_n(G_{l,r}^{(n-1)}, u_{l,r}^{(n-1)}) + rM_n(G_{l,r}^{(n-1)}, w_{l,r}^{(n-1)})))$. By (b.1), $F_n(l(M_n(G_{l,r}^{(n-1)}, u_{l,r}^{(n-1)}) + rM_n(G_{l,r}^{(n-1)}, w_{l,r}^{(n-1)}))) = F_n(g_1(l, r), \ldots, g_q(l, r))$ where $g_i \in F$ for some finite set $F$ of functions of the form $f(l, r) = \sum_{i=0}^{n+1} \sum_{j=0}^{i} \sum_{x=0}^{n} \sum_{y=0}^{x} a_{i,j,x,y} l^i r^j \frac{r^y}{(1+r)^x}$. By Lemma C.2 then, using a line of proof similar to (b.1), we have that $\forall j \in [q] \; F_n(l(M_n(G_{l,r}^{(n-1)}, u_{l,r}^{(n-1)}) + rM_n(G_{l,r}^{(n-1)}, w_{l,r}^{(n-1)})))_j$ is one of finitely many possible functions of the aforementioned form.

b.3 If $RO_i = avg$ then $R_n(M_n(G_{l,r}^{(n-1)})) = F_n(\frac{l(M_n(G_{l,r}^{(n-1)}, u_{l,r}^{(n-1)}) + rM_n(G_{l,r}^{(n-1)}, w_{l,r}^{(n-1)}))}{l(1+r)}) = F_n(\frac{M_n(G_{l,r}^{(n-1)}, u_{l,r}^{(n-1)}) + rM_n(G_{l,r}^{(n-1)}, w_{l,r}^{(n-1)})}{1+r})$. Using a line of proof similar to (b.2), we have

that $\forall j \in [q]$ $F_n\big(\frac{M_n(G_{l,r}^{(n-1)}, u_{l,r}^{(n-1)}) + r M_n(G_{l,r}^{(n-1)}, w_{l,r}^{(n-1)})}{1+r}\big)_j$ is one of finitely many possible

functions of the form $f(l,r) = \sum_{i=0}^{n} \sum_{j=0}^{i} \sum_{x=0}^{n+1} \sum_{y=0}^{x} a_{i,j,x,y} l^i r^j \frac{r^y}{(1+r)^x}$.

c) For the final readout $R$, whether $R = sum$ or $R = avg$, by the induction assumption and using a line of proof similar to (b.3), we have that $\forall j \in [r]$ $R(G^{(m)})_j$ is one of finitely many possible

functions of the form $\sum_{i=0}^{m+1} \sum_{j=0}^{i} \sum_{x=0}^{m+1} \sum_{y=0}^{x} a_{i,j,x,y} l^i r^j \frac{r^y}{(1+r)^x}$. $\qquad\square$

**Lemma 4.6.** *For $l \in \mathbb{N}_{>0}, r \in \mathbb{N}_{>0}$, define $h(G_{l,r}) := l\big(\frac{3+2re^9}{1+re^9} + r\frac{3+2re^{12}}{1+re^{12}}\big)$.*

*Let $f(l,r) = \sum_{i=0}^{m+1} \sum_{j=0}^{i} \sum_{x=0}^{m+1} \sum_{y=0}^{x} a_{i,j,x,y} l^i r^j \frac{r^y}{(1+r)^x}$ for some coefficients $\{a_{i,j,x,y}\}$, then*

$$\exists r_0 : \forall r > r_0 \; \lim_{l \to \infty} |f(l,r) - h(G_{l,r})| = \infty$$

*Proof.* 1.  If there is $i > 1$ such that $a_{i,j,x,y} \neq 0$ for some $j, x, y$, then clearly $\lim_{l \to \infty} |f(l,r) - h(G_{l,r})| = \infty$ for every $r$, and same if there exist no $i > 0$ and $j, x, y$ for which $a_{i,j,x,y} \neq 0$. This is because the function $h(G_{l,r})$ is linear in $l$.

2. Otherwise, we can assume that

$$f(l,r) = \sum_{i=0}^{1} \sum_{j=0}^{i} \sum_{x=0}^{m+1} \sum_{y=0}^{x} a_{i,j,x,y} l^i r^j \frac{r^y}{(1+r)^x}$$

Define

$$f_1(l,r) := \sum_{x=0}^{m+1} \sum_{y=0}^{x} \big(\frac{a_{0,0,x,y}}{l}\big) r^j \frac{r^y}{(1+r)^x}, \quad f_2(l,r) := \sum_{j=0}^{1} \sum_{x=0}^{m+1} \sum_{y=0}^{x} a_{1,j,x,y} r^j \frac{r^y}{(1+r)^x}$$

Then, we have that $f(l,r) = l(f_1(l,r) + f_2(l,r))$ and

$$|f(l,r) - h(G_{l,r})| = l \left| f_1(l,r) + f_2(l,r) - \big(\frac{3+2re^9}{1+re^9} + r\frac{3+2re^{12}}{1+re^{12}}\big) \right|$$

Note that $\lim_{l \to \infty} f_1(l,r) = 0$; $f_2$ is independent of $l$; and since $f_2$, $r \mapsto \big(\frac{3+2re^9}{1+re^9} + r\frac{3+2re^{12}}{1+re^{12}}\big)$ are distinct rational functions of $r$ they can only coincide at finitely many values of $r$. Hence, $\exists r_0 : \forall r > r_0 \; \lim_{l \to \infty} |f(l,r) - h(G_{l,r})| = \infty$. $\qquad\square$

**Theorem 4.7.** *No MPGNN+VN can approximate $h$ for all graphs $\{G_{l,r}\}$. Formally, MPGNN+VNs $\not\approx h$.*

*Proof.* By Lemma 4.5 we have that no matter the $l, r$, $B(G_{l,r})$ is necessarily one of finitely many functions $\{f_1, \ldots, f_k\}$ such that, by Lemma 4.6, $\forall i \in [k] \exists r_i : \forall r > r_i \; \lim_{l \to \infty} |f_i(G_{l,r}) - h(G_{l,r})| = \infty$. Hence, $\forall r > max(r_i : i \in [k]) \; \lim_{l \to \infty} |h(G_{l,r}) - B(G_{l,r})| = \infty$. $\qquad\square$

**Lemma 4.8.** *There exists a GPS that computes $h$ exactly.*

*Proof.* Define an attention head $H = (W_K, W_Q, W_V)$ as follows:

- $W_Q = ((1, 1))$

- $W_K = ((2, 3))$

- $W_V = ((2, -1))$

We have that

$$\text{softmax}(Z(G_{l,r})W_Q(Z(G_{l,r})W_K)^T)Z(G_{l,r})W_V = \text{softmax}\left(\begin{pmatrix} 21\cdots 21 & 30\cdots 30 \\ 28\cdots 28 & 40\cdots 40 \end{pmatrix}\right)(3\cdots 3 \quad 2\cdots 2)^T =$$

$$\left(\frac{3+2re^9}{1+re^9}\cdots\frac{3+2re^9}{1+re^9} \quad \frac{3+2re^{12}}{1+re^{12}}\cdots\frac{3+2re^{12}}{1+re^{12}}\right)^T$$

That is,

$$H(u_i)_{l,r} = \frac{l3e^{21}+lr2e^{30}}{le^{21}+lre^{30}} = \frac{3e^{21}+2re^{30}}{e^{21}+re^{30}} = \frac{3+2re^9}{1+re^9}$$

$$H(w_i)_{l,r} = \frac{l3e^{28}+lr2e^{40}}{le^{28}+lre^{40}} = \frac{3e^{28}+2re^{40}}{e^{28}+re^{40}} = \frac{3+2re^{12}}{1+re^{12}}$$

It is not difficult to verify that there is a GPS $B$ comprising attention $H$ and a final sum readout, such that $B(G_{l,r}) = l(H(u)_{l,r} + rH(w)_{l,r}) = l(\frac{3+2re^9}{1+re^9} + r\frac{3+2re^{12}}{1+re^{12}}) = h(G_{l,r})$. □

**Corollary 4.9.** *MPGNN+VNs* $\not\succeq^{\{1,2\}}$ *GPS.*

*Proof.* By Theorem 4.7 no MPGNN+VN can approximate $h$, and by Lemma 4.8 there exists a GPS that computes $h$ exactly. □

Table 2: Hyperparameters for GatedGCN+VN.

|  | PascalVOC-SP | COCO-SP | Peptides-Func | Peptides-Struct | ogbg-code2 | ogbg-molpcba |
|---|---|---|---|---|---|---|
| lr | 0.001 | 0.001 | 0.001 | 0.001 | 0.001 | 0.001 |
| dropout | 0.1 | 0.1 | 0.1 | 0.1 | 0.3 | 0.4 |
| #layers | 10 | 6 | 8 | 10 | 5 | 6 |
| hidden dim. | 70 | 90 | 80 | 70 | 256 | 1024 |
| head depth | 2 | 1 | 2 | 2 | 1 | 1 |
| PE/SE | none | none | RWSE | LapPE | none | RWSE |
| VN pooling | mean | mean | mean | mean | mean | sum |
| batch size | 50 | 50 | 200 | 200 | 32 | 512 |
| #epochs | 200 | 200 | 250 | 250 | 50 | 60 |
| #Param. | 478K | 453K | 486K | 466K | 1.20M | 57M |
| time/epoch | 66s | 206s | 24s | 28s | 502s | 487s |

Table 3: Hyperparameters for GCN+VN.

|  | PascalVOC-SP | COCO-SP | Peptides-Func | Peptides-Struct |
|---|---|---|---|---|
| lr | 0.001 | 0.001 | 0.001 | 0.001 |
| dropout | 0.2 | 0.1 | 0.1 | 0.1 |
| #layers | 8 | 10 | 10 | 8 |
| hidden dim. | 105 | 95 | 95 | 105 |
| head depth | 2 | 1 | 3 | 2 |
| PE/SE | none | none | RWSE | LapPE |
| VN pooling | mean | mean | mean | mean |
| batch size | 50 | 50 | 200 | 200 |
| #epochs | 200 | 200 | 250 | 250 |
| #Param. | 460k | 465k | 487k | 474k |
| time/epoch | 35s | 156s | 18s | 19s |

# D  EXPERIMENT DETAILS

## D.1  DATASETS

The LRGB dataset collection (Dwivedi et al., 2022) has recently been introduced as a set of benchmarks that rely on long-range interactions and are thus especially well-suited to benchmark architectures that can handle long-range interactions such as graph transformers. Here, we use 4 LRGB datasets: peptides-func and peptides-struc, which are graph-level classification and regression tasks, respectively, as well as the two vision datasets Pascal-VOC and COCO, which both model semantic segmentation on images as node classification tasks on superpixel graphs. Very recently, Tönshoff et al. (2023) showed that the original baseline results reported by Dwivedi et al. (2022) and Rampášek et al. (2022) suffered from suboptimal hyperparameters and lack of feature normalization. We will use their updated baseline values in our experiments.

Additionally, we incorporate two graph-level datasets from the OGB project (Hu et al., 2020a): ogbg-code2 aims to predict the name of python methods presented as abstract syntax trees. The molecular dataset ogbg-molpcba aims to regress the activity of molecules for 128 bioassays.

## D.2  HYPERPARAMETERS

For our experiments on LRGB we adopted the hyperparameter ranges and tuning methodology from Tönshoff et al. (2023) and strictly adhere the the parameter budget of 500K. For the OGB datasets we train larger models as no official parameter budgets are given. Note that we view the usage of positional or structural encodings as a hyperparameter that is either chosen as LapPE (Rampášek et al., 2022), RWSE Dwivedi et al. (2021) or "none". On ogbg-molpcba in particular the usage of RWSE has a significant positive effect on performance. Table 2 and Table 3 provide the chosen final hyperparameters and runtimes per epoch for GatedGCN and GCN, respectively. Note that in the final runs we average the results over 4 random seeds for LRGB datasets and over 10 random seeds for OGB datasets.

### D.3 SYNTHETIC DATA GENERATION

For our synthetic experiments we use random Erdős-Rényi graphs with a mean degree samples uniformly from the interval $[1, 5]$. Note that we also use this graph distribution to generate the graphs $G_{l,r}$ from Section 4.3 even though the formal definition specifies an empty edge set. This does not affect the theoretical ability of GPS to express the target function $h(G_{l,r})$ but it does allow us to test if the model can learn to ignore the edges in a data driven manner. For both experiments, we train on 100K randomly generated graphs and test on the parameter ranges specified in Section 5.1.

