# OpenReview forum: "Distinguished In Uniform: Self-Attention Vs. Virtual Nodes"
_ICLR.cc/2024/Conference — ICLR 2024 poster_

### Official Review · Reviewer_MBi3 · 2023-10-19

**Soundness:** 3 good
**Presentation:** 4 excellent
**Contribution:** 3 good
**Rating:** 6
**Confidence:** 2

**Summary:**

This paper studies the expressiveness of graph transformers and message-passing gnns. As its first step, the authors exploit the universality of MLP and the property of positional encoding, i.e., injective up to isomorphism, to prove the non-uniform universality of graph transformer and message-passing gnns. Then, the scope of discussion is extended to uniform expressivity, that is, whether these kinds of neural architectures can approximate arbitrary function no matter how large the input graph is. Basically, the authors show that both graph transformers and message-passing gnns are not universal approximator in this setting. Moreover, they offer important insight that these two kinds of neural architectures do not subsume each other. Specifically, graph transformers cannot perform unbounded aggregation, yet its attention mechanism allows asymmetric weighting of incoming messages. Accordingly, the authors design and conduct experiments on synthetic datasets to validate their theoretical results. On practical datasets, these models are also comparable, especially the virtual node trick, which helps a lot for message-passing gnns. In summary, this paper tells the community that attention is NOT all you need in graph learning.

**Strengths:**

1.	This paper is well-written. I can effortlessly pick the main points up.
2.	The theoretical results introduced in this paper seem to be crucial for developing machine learning models dedicated to graph data. Notably, these results explain some interesting phenomena emerging in recent years, including the superiority of graph transformers in some competitions, the surprising usefulness of virtual node trick, and the existence of some real-world datasets on which message-passing gnns are still state-of-the-art.
3.	The experiments are convincing, where the difference between these two kinds of neural architectures is remarkable.

**Weaknesses:**

1.	It seems that the presented theoretical results in the non-uniform are relatively trivial, as they are straightforward results of the combination of MLP’s universality and PE’s discrimination capacity.
2.	The difference and respective advantages deserve to be connected to practical tasks on molecular graphs, as there have been many public tasks, some of which graph transformers outperform traditional message-passing gnns, yet some are not. Such connections must be helpful for the community and make the theoretical results practical.

**Questions:**

In the synthetic experiments, the authors said they were interested in the generalization behavior of the train models. However, the setting is not as usual. It is not an i.i.d. generalization but o.o.d. extrapolation (graph size <= 50 during training and > 50 in test). Generally and intuitively, a model that is more sophisticated with o.o.d. extrapolation is often due to its more reasonable inductive bias or limited hypothesis space, such that it captures the underlying actual mapping rather than fitting the training data by other consistent yet different mappings. Thus, I need clarification about the rationale behind experimental design. Could you explain this to me?

---

> ### Author Response · Authors · 2023-11-16
> **Author Response**
>
> We thank the reviewer for the insightful comments. Please find below our response to your question:
>
> By "generalization" we mean correctly handling graphs of sizes larger than seen during training, which is indeed generalization in the o.o.d. sense. We hypothesize that the latter corresponds with uniform expressivity i.e.:
> 1) That uniform inexpressivity implies an extremely low probability of learning a GNN that generalizes (to o.o.d. graph sizes).
> 2) That there is a good probability of learning uniformly expressive networks (which by definition generalize) when they exist.
>
> If our hypothesis is true, then differences (between architectures) in uniform expressivity are meaningful in practice, making theoretical results regarding uniform expressivity practically relevant. Our synthetic-data experiments are meant to examine the above:
> 1) To test the magnitude of the generalization error of a learned GNN when the architecture is uniformly inexpressive of the target function.
> 2) To test how well we can learn from data (using SGD) a well-generalizing GNN, when a uniformly expressive GNN is known to exist.
>
> We will try to add clarification regarding the above, in the updated manuscript.

---

> > ### Comment · Reviewer_MBi3 · 2023-11-21
> > **Response**
> >
> > Thanks for your detailed explanation. I will keep the rating unchanged.

---

> > > ### Author Response · Authors · 2023-11-22
> > > **Author Response**
> > >
> > > Let us additionally comment on the weaknesses pointed out by the reviewer in addition to answering the explicit questions.
> > >
> > > **Q**: Weaknesses: "It seems that the presented theoretical results in the non-uniform are relatively trivial..." \
> > > **A**: It is true that the proofs in the non-uniform setting are rather straightforward as they effectively restate a known proof, showing that it can be applied to a variety of settings (and not just graph transformers). It is therefore not the main technical contribution of the paper. We included this section for two reasons:
> > > 1. They complete the formal picture and provide a contrast to the uniform setting results.
> > > 2. They clarify some prior results from the graph transformer literature.
> > > Some works, such as GPS, highlight the fact that their approach can universally approximate functions in a non-uniform setting as an important property of their method without clarifying that this is the case for virtually any architecture when using the same positional encoding.
> > > We believe it is important to emphasize formally that this somewhat trivial result is not a unique property of graph transformers and therefore certainly not an advantage.
> > >
> > >
> > > **Q**: Weaknesses: "The difference and respective advantages deserve to be connected..."\
> > > **A**: We do not have a clear recommendation to use one or the other model for this or that specific task. Our results convey a general message: Neither GPS nor MPGNN+VN performs strictly better than the other across all learning tasks. We believe that this message, although general, is relevant to practice: It can increase awareness among practitioners that they should consider both models when tackling a learning task instead of believing one model to be superior and dismissing the other without additional thought.\
> > > Moreover, our results do offer a general connection: If the target function is estimated to include the global sum of some data of the nodes then MPGNN+VN may be the better choice to start with, and if the target function is estimated to include a sophisticated global weighted average where the weight depends on both sides then GPS may be the better choice.

---

### Official Review · Reviewer_EnRw · 2023-10-28

**Soundness:** 3 good
**Presentation:** 3 good
**Contribution:** 3 good
**Rating:** 8
**Confidence:** 4

**Summary:**

Summary: This work can be contextualized along a recent line of study in graph learning which is focused on comparing graph transformers (GTs) and message passing GNNs (MPGNNs) and finding out which is better and why. This paper in particular theoretically and empirically compares the expressive power of GTs and MPGNNs with virtual nodes (MPGNN+VNs) in the uniform setting where a single model must work for graphs of all sizes. It shows that neither model is uniformly universal, but they can express some unique functions, making their expressive power incomparable.

the paper's contributions:
- Presents important insights that can be useful to understand the working capabilities of GTs and MPGNNs.
- Proves that GTs and MPGNN+VNs cannot uniformly approximate every computable graph function, even with polynomial-time positional encodings.
- Shows GTs cannot uniformly approximate unbounded summation like |V|2, while MPGNN+VNs can.
- Shows MPGNN+VNs cannot uniformly approximate functions exploiting softmax attention's asymmetric neighbor weighting, while GTs can.

**Strengths:**

Strengths:
- The paper makes an important theoretical contribution by formally proving distinguishing functions for GTs and MPGNN+VNs. This helps characterize their expressive power, and informs more about the strengths and weaknesses of these two model class in addition to what is known in recent literature (eg Cai et al., 2023).
- The proofs identifying unique functions are non-trivial and provide insight into the core operations enabling GTs and MPGNN+VNs to express different functions.
- The theoretical findings are verified through special designed experiments on synthetic data, showing the distinguishing functions are learnable in practice.
- Experiments on real-world benchmarks demonstrate MPGNN+VNs can be competitive with GTs in some cases, due to global communication via virtual nodes. however, this is known in the literature, to the best of my understanding

**Weaknesses:**

Limitations and Questions:
- The theoretical analysis focuses on comparing one variant of GTs (GPS) and MPGNN+VNs. Results could vary for different architectures within these families. How accurate would this generalization be?
- On some realworld datasets, MPGNN+VNs do not fully close the performance gap compared to GTs. It is unclear if this limitation is fundamental or if deeper MPGNN+VNs could match GTs.

**Questions:**

in the Weaknesses section

---

> ### Author Response · Authors · 2023-11-16
> **Author Response**
>
> We thank the reviewer for the thorough review. Below, we comment on the two weaknesses pointed out in the review.
>
> **Q1**: The theoretical analysis focuses on comparing one variant of GTs (GPS) and MPGNN+VNs. Results could vary for different architectures within these families. How accurate would this generalization be?\
> **A1**: The first question regards the families of MPGNN+VN and GT architectures to which our theory applies.
> Formally reasoning about these architectures naturally requires us to fix some specific design choices, but we would argue that both GPS and MPGNN+VN represent very large and practically relevant classes of models.
> In particular, our theory is completely agnostic to the choice of the message passing module and holds true for any popular MPGNN layer such as GCN, GIN, GAT, or GatedGCN.
> This is clear when looking at the graphs we use to prove the differences because they both contain no edges so no messages are sent during the message passing step.\
> The intuition underlying Theorem 4.3 and Corollary 4.4 is that weighted averages like softmax self-attention struggle to uniformly approximate functions that depend on the absolute number of vertices.
> We expect that this generalize to most graph transformers with global information exchange built on such weighted averages.
> Of course, the formal proof details may need to be adapted for GTs outside of the GPS framework.
> We will update the camera-ready version to make this part clearer.
>
> **Q2**: On some realworld datasets, MPGNN+VNs do not fully close the performance gap compared to GTs. It is unclear if this limitation is fundamental or if deeper MPGNN+VNs could match GTs.\
> **A2**: This is a broad, but highly important question, we are also very interested in. At least in our experiments, going deeper (while keeping the parameter limits of LRGB) did not lead to better results. Most notably, there is a gap on the vision dataset COCO where pure vision transformers achieve even better results. Nevertheless, it is an intriguing question what aspects are key to the vastly different performance of the various graph learning models on that dataset.

---

### Official Review · Reviewer_BMpU · 2023-10-31

**Soundness:** 2 fair
**Presentation:** 2 fair
**Contribution:** 3 good
**Rating:** 6
**Confidence:** 3

**Summary:**

This paper comprehensively compares the expressivity of Graph Transformers and Message-Passing GNNs.

This paper presents the following conclusions:

1) Neither Graph Transformers nor Message-Passing GNNs are universal in the uniform setting.

2) There are functions that Graph Transformers can express while Message-Passing GNNs with virtual nodes can not.

3) Even with perfect positional encoding, the expressiveness of Graph Transformers and MPGNNs with virtual nodes differs substantially.

This paper conducts experiments on real data and synthetic data to verify the theoretical analysis proposed in the paper.

---- After rebuttal---
Thanks for the authors' response. My main concern is strong assumption of the uniform setting and the experimental results.

For the strong assumption of the uniform setting, the author explained that
﻿the uniform setting is a good representation of scenarios because the graph sizes at
training time are smaller than the graph sizes during inference. I generally agree with the author's response.

For the novelty. The author reclaimed their novelty, which is not summarized (or even mentioned) in the introduction. Now, the author summarize this paper's novelty about the theoretical provements and the proposed synthetic data. I agree these two thing are new. However, the experimental results on real data still exists. The authors can carefully fix the minor mistakes or typos in their revised version.

Given the promise that the author will add the detailed proof and more analysis on the real dataset, I raise the score from reject to broadline accept.

**Strengths:**

1) This paper relates two common designs in graphs, i.e., graph transformer and message-passing GNN.

2) This paper gives a theoretical analysis of the capabilities	of two basic GNN designs and conducts comprehensive experiments to verify the theoretical analysis.

3) The study provides insightful results that add depth to our understanding of the expressiveness of Graph Transformers and MPGNNs in scenarios with optimal positional encoding.

4) The differentiation in the capabilities of Graph Transformers and MPGNNs is well highlighted, offering clarity on their respective strengths and limitations.

**Weaknesses:**

The important concern is the writing. Although this is primarily a theoretical paper, it does not express its flow of proof clearly. I recommend the author to improve their writing.

1) The most important weakness of this paper is writing. The introduction is not easy to understand.

2）**Lack of Justification**: The paper focuses on scenarios where positional encoding is injective. However, there is a noticeable lack of justification for why this particular scenario is important or realistic. To ensure that the results derived hold value in practical applications, it is essential to provide a clear context and relevance for the chosen scenario.

3) The process of proof is not so clear. For example, in Section 4.2, the author attempts to prove GPS do not subsume MPGNN+VNs. However, I can not understand the main path of proof of Theorem 4.3 and Corollary 4.4.

4) The strong assumption of uniform setting made in the paper may not align with the practical case.

5) **Redundancy in Experimental Results**: Similar experimental outcomes have been presented in multiple prior works, notably:
    - Tönshoff, Jan, et al. "Where did the gap go? Reassessing the long-range graph benchmark." arXiv preprint arXiv:2309.00367 (2023).
    - Cai, Chen, et al. "On the connection between MPNN and graph transformer." arXiv preprint arXiv:2301.11956 (2023).
   While building upon prior work is a hallmark of research progression, it's crucial to ensure that the presented findings either provide a novel perspective or build significantly upon the existing literature.

Given the value of the derived results, this paper can contribute substantially to the field with some revisions. I recommend a clear justification for the chosen scenario of positional encoding and a distinction between the presented findings and existing literature. This would fortify the paper's novelty and relevance, making it a more significant contribution to the domain.

**Questions:**

N/A

---

> ### Author Response · Authors · 2023-11-16
> **Author Response**
>
> We thank the reviewer for the constructive comments.
> Please find below our clarifications and responses to some of your comments.
>
> **Q**: Weaknesses (1): "The most important weakness of this paper is writing. ..."\
> **A**: We agree that the writing can be improved throughout the manuscript and will address this in the revised version. We will also update the paper with a clearer introduction.
>
> **Q**: Weaknesses (2): "the paper focuses on scenarios where positional encoding is injective..."\
> **A**: We consider injective positional encoding only for section 3, where we extend known results in the non-uniform expressivity setting. The main focus of the paper is the uniform expressivity setting, discussed mainly in section 4. Our results there are *not limited to* having (injective) positional encodings. Rather, the inexpressivity results for GPS hold *even when* (pg 7, lines 1;4;8) the initial graph is featured with positional encodings.
> We will review the presentation to see how these points can be made clearer in the camera-ready version.
>
> **Q**: Weaknesses (3): "the process of the proof is not so clear..."\
> **A**: We assume that you refer to the proof overview (in the main part), we agree that it should be made clearer, and we intend to do so in the camera-ready version.
>
> **Q**: Weaknesses (4): "The strong assumption of uniform setting..."\
> **A**: We argue that the uniform setting is highly relevant to practice for the following reasons:
> As demonstrated in our experiments, the uniform setting is a good representation of scenarios where the graph sizes at training time are smaller than the graph sizes during inference. In practice, it is desirable to learn models that robustly infer missing labels on data outside of the training distribution.
> Furthermore, due to compute-resource limitations and relevancy requirements, it may be desired not to re-train whenever new graphs are larger than seen before e.g. when the system modeled by the graphs is growing over time.
> We agree that a justification like the above should be mentioned explicitly in the paper and we intend to add it in the camera-ready version.
>
> **Q**: Weaknesses (5): "Redundancy in Experimental Results..."\
> **A**: First, we would like to clarify that the experimental part of this work is meant to complement the theoretical part, it is *a* contribution but not the main contribution.
> The core contributions of our experiments are two-fold:
> 1) Our study on synthetic data is new. It provides the first direct empirical comparison of graph transformers and virtual nodes with regard to uniform function approximation.
> 2) Our experiments on real-world data provide a thorough evaluation of MPGNNs with virtual nodes on LRGB and other benchmark datasets.
> There is some overlap with the experimental setup of Cai et al., but we should point out that their models do not adhere to the official 500k parameter budget and are comparatively under-tuned, as our configurations yield better results with smaller models.
> Tönshoff et al. identify some weaknesses in the LRGB results reported for MPGNNs and provide improved values, but only for models without VN.
> Our experiments extend their methodology to MPGNNs *with* VN to obtain novel and improved results for this class of models.
>
> We will improve the presentation of the experiments in the updated manuscript to highlight this more clearly.

---

> > ### Comment · Reviewer_BMpU · 2023-11-22
> >
> > Thanks for the reply. I would like to raise some follow-ups:
> >
> > **On the Choice of Positional Encoding in Uniform vs. Non-Uniform Settings:**
> > Could you elaborate on the decision to utilize injective positional encoding in the uniform setting and PTIME positional encoding in the non-uniform setting? What is the theoretical or practical significance of applying different strengths of positional encoding in these distinct scenarios?
> >
> > **Comparative Strength of Positional Encodings:**
> > It appears that injective positional encoding is more robust compared to PTIME positional encoding. Can you clarify whether this is indeed the case, and if so, why the stronger encoding was not applied to the more challenging non-uniform case?
> >
> > **Impact on Expressiveness and Generalization:**
> > How does the choice of positional encoding impact the expressiveness and generalizability of the graph transformers and message-passing GNNs in practical applications? Is there a trade-off that is being optimized by using injective encoding in one scenario over the other?
> >
> > **Consistency in Methodology:**
> > Would the results and conclusions presented in the paper benefit from a consistent application of positional encoding across both settings? Could this potentially highlight differences in expressiveness due to the model architecture rather than the encoding strength?

---

> > > ### Author Response · Authors · 2023-11-23
> > > **Author Response**
> > >
> > > We thank the reviewer for the follow-up response. Let us clarify the questions raised in the following.
> > > In the camera-ready manuscript, we will strengthen the clarity with which the points below are presented.
> > >
> > > First, we would like to clarify that the goal of the paper is not to compare different PEs nor to recommend what PEs to use in which scenarios and with which models.
> > > Rather, the goal is to compare the models themselves while taking into account PEs that go along with such models in the relevant literature and practice.\
> > > In Section 3 we consider injective PEs because they are used in the literature to achieve universal approximation.\
> > > In Section 4.1 we mention PTIME PEs as something that *does not help* to achieve universality (in the uniform setting).\
> > > In Sections 4.2 we mention bounded PEs as something that *does not help GPS* express the target function.
> > >
> > > **Choice and Comparative Strength:**\
> > > Overall there are three different conditions on positional encodings that are relevant throughout our results:
> > > 1. Injectiveness
> > > 2. Computational time-complexity
> > > 3. Boundedness
> > >
> > > There is no strict hierarchy between these conditions and neither implies the other, but there are principled reasons for the specific conditions chosen for each result as discussed in the following.
> > >
> > > First, let us emphasize that we use injective PEs in the **non**-uniform setting, not the uniform setting. This is in line with prior works which mostly consider this non-uniform setting and provide analog proofs using the same assumption.
> > > Here the injectivity is needed to allow for the straightforward proof of universality based on memorizing the correct output for each unique graph.
> > > This proof logic can not be translated to a uniform setting of unbounded graph size and the strong assumption of injectiveness is therefore not used in Section 4.
> > > Moreover, PEs commonly used in practice, such as LapPE, are also simply not injective on graphs of unbounded size, which also is a practical reason for dropping this condition.
> > >
> > > A common characteristic of PEs that does hold in the uniform setting is their polynomial computation and boundedness.
> > > Hence, we build our theory on uniform expressivity around these assumptions, which also hold when using popular structural encodings such as RWSE or no encoding at all.
> > > To get the strongest possible version of each theorem we only use the specific assumption necessary for each proof, which is why we only assume PTIME computibility in Section 4.1 and only boundedness in Section 4.2.
> > > We could require additional properties from the PEs in both sections, but this would only yield weaker statements.
> > >
> > > **Consistency in Methodology:**\
> > > We claim that the PEs are applied consistently to both models, and that we do highlight differences or indifferences between the model architectures themselves:\
> > > In Section 3 we consider injective PEs for both models.\
> > > In Section 4.1 we consider PTIME PEs for both models.\
> > > In Section 4.2 we show that when GPS has bounded PEs at its disposal, it is less expressive than MPGNN+VN without them, for the target function we define there. Having PEs (in addition to the initial node features) can only improve the expressivity of a model, and not having them can only worsen it. Hence, it is immediate that a consistent application of the PEs would yield the same result: If GPS will not have PEs then of course it still cannot express the target function, and if MPGNN+VN will have PEs then of course it still can express the function.
> > >
> > > **Impact of PEs:**\
> > > PEs can have a significant impact on many graph learning tasks.
> > > There are even some graph transformers where PEs are the only way the transformer may access the underlying graph structure.
> > > Even in GPS where message passing and transformer attention happen in parallel, PEs have strong practical implications, often improving experimental results significantly (but that is again dataset dependent).
> > > There are several datasets (e.g. ogbg-molpcba), for which using RWSE instead of the injective LapPE gives better results (consistently over MPGNN and GT architectures), so injectivity is not always crucial in practical applications.\
> > > But we would like to emphasize again that the main focus of our paper is not to recommend specific encodings but to compare the expressiveness of different architectures.
> > >
> > > **Uniform vs non-uniform expressivity, strength:**\
> > > We would like to clarify that the uniform setting is the stronger, or "more challenging", form of expressivity since a uniformly expressing model is also non-uniformly expressing.
> > > This is an immediate consequence of the definition.

---

### Official Review · Reviewer_Uru7 · 2023-11-07

**Soundness:** 3 good
**Presentation:** 3 good
**Contribution:** 3 good
**Rating:** 6
**Confidence:** 3

**Summary:**

This paper compares model expressivity between graph transformers (GT) and massage passing GNNs with virtual nodes (MPGNN+VN). The authors theoretically demonstrate that GT and MPGNN+VN are universal function approximators on the graph under uniform setups, where different neural networks can be utilized for every graph size. However, under the non-uniform case where one neural network is supposed to work for all the graphs, both are not universal approximators and express different sets of functions, indicating they do not subsume each other. The authors also conduct numerical experiments to validate the theoretical findings.

**Strengths:**

First of all, thank the authors for their submission to ICLR. This paper offers valuable insights into the expressive power of graph neural networks. Specifically, it exhibits the following notable strengths:
(1)	The authors provide a novel and insightful comparison between Graph Transformer and MP-GNN (with VN) in both non-uniform and uniform setups. The latter setup, in particular, marks the first time it has been explored from this perspective, revealing that GT and MPGNN-VN do not subsume each other.
(2)	The paper includes numerical experiments that complement the theoretical insights. The authors also attach the code, which will benefit the community.
(3)	Last but not least, the paper is well organized, and the writing is straightforward, so it is easy to follow even though the underlying proof is remarkable.

**Weaknesses:**

Despite the strengths listed above, there are still areas in the paper where improvements can be made to enhance clarity and overall significance.
(1)	Regarding the writing:
a.	The paper utilizes many acronyms, which may necessitate repeated explanations. For instance, in Figure 1, it would be helpful if the author could reiterate the meanings of "MP," "VN," “SA,” “PH,” and "FF" in the figure caption. Additionally, in Figure 2(b), clarifying the definitions of "l" and "r" would enhance writing clarity. It is also recommended that the authors create a table in the appendix with all the acronyms.
b.	The utilization of some math symbols may confuse and misleading. For example, in subsection 4.3, the “lr” represents “l*r” instead of learning rate or number of layers.
c.	There are some missing values in Table 1, which may be better to illustrate the reason in the table caption in addition to the other place.
d.	Some minor typos and grammars, such as “Based on the the positional encoding LapPE”.
(2)	Regarding the theory and numerical experiments:
a.	The assumptions and limitations of the proposed theory are somewhat unclear. For instance, it is not clearly defined whether the theory is applicable to various graph tasks or solely focused on graph classification tasks.
b.	The message conveyed in Section 5.2 is also vague and disconnects with previous sections. Given that Table 1 indicates that the best performance appears somewhat random, it is not convincing how the theoretical insights can guide the practice. It may be valuable to conduct a more thorough investigation into the utilization of theoretical insights for model selection based on the characteristics of the dataset.

**Questions:**

The questions are mainly related to the “weakness”:
(1)	Please add a list of math symbols and abbreviations in the appendix and clarify the above writing questions.
(2)	Does the theory also satisfy different graph learning tasks?
(3)	How can we utilize theory to understand the results of section 5.2?

---

> ### Author Response · Authors · 2023-11-16
> **Author Response**
>
> We thank the reviewer for the constructive comments, we will carefully revise our paper taking them into account.
> Please find below our clarifications and responses to some of your comments.
>
> **Q**: Weaknesses (1): "Regarding the writing..."\
> **A**: We agree with the suggestions and will implement them in the manuscript.
>
> **Q**: Weaknesses (2a): "The assumptions and limitations of the proposed theory are somewhat unclear..."\
> **A**: Our theoretical results concern graph-level function-approximation tasks. Some of the results can be extended to node-level tasks without much effort, while for other results more work is required to make conclusions concerning node-level tasks.
> For example, it is not difficult to create a node-level function that GTs cannot learn while MPGNN+VNs can (generalizing Corollary 4.4).
> We will add a clarification to the manuscript.
>
> **Q**: Weaknesses (2b): "The message conveyed in Section 5.2 is also vague..."\
> **A**: Unlike the synthetic data experiments, our experiments on real-world problems are not meant to demonstrate our theoretical results directly. Rather, they offer another view leading to the general message of the paper: Neither GPS nor MPGNN+VN performs strictly better than the other across all learning tasks. Hence, the first conclusion from both our theoretical and experimental results is that both models should be considered in the general case. However, the theoretical insights suggest that if the target function is estimated to include the sum of some data of the nodes then MPGNN+VN may be the better choice to start with, and if the target function is estimated to include sophisticated weighted averages where the weight depends on both sides then GPS may be the better choice.
> We will try to add a clarification in the manuscript.

---

### Meta-Review · Area_Chair_pTkh · 2023-12-04

**Metareview:**

This paper compares how expressive graph transformers and GNNs are.  The paper gives results on how theoretically they are universal function approximaters. The reviewers felt that this theoretical contribution gives insight into neural networks.  Some weaknesses were pointed out in some of the results, but overall the reviewers felt the paper was of interest to the community.

**Justification For Why Not Higher Score:**

The paper presents some good results that are of interest, but some of the results could have been stronger.  I think the paper is fit for acceptance, but not a spotlight.

**Justification For Why Not Lower Score:**

There was enough interest in the theoretical results that our community will benefit from this publication.

---

### Decision · Program_Chairs · 2024-01-16

Accept (poster)